# Learning from Delayed Feedback in Games via Extra Prediction

**Yuma Fujimoto**
CyberAgent
fujimoto.yuma1991@gmail.com

**Kenshi Abe**
CyberAgent
abe_kenshi@cyberagent.co.jp

**Kaito Ariu**
CyberAgent
kaito_ariu@cyberagent.co.jp

## Abstract

This study raises and addresses the problem of time-delayed feedback in learning in games. Because learning in games assumes that multiple agents independently learn their strategies, a discrepancy in optimization often emerges among the agents. To overcome this discrepancy, the prediction of the future reward is incorporated into algorithms, typically known as Optimistic Follow-the-Regularized-Leader (OFTRL). However, the time delay in observing the past rewards hinders the prediction. Indeed, this study firstly proves that even a single-step delay worsens the performance of OFTRL from the aspects of social regret and convergence. This study proposes the weighted OFTRL (WOFTRL), where the prediction vector of the next reward in OFTRL is weighted $n$ times. We further capture an intuition that the optimistic weight cancels out this time delay. We prove that when the optimistic weight exceeds the time delay, our WOFTRL recovers the good performances that social regret is constant in general-sum normal-form games, and the strategies last-iterate converge to the Nash equilibrium in poly-matrix zero-sum games. The theoretical results are supported and strengthened by our experiments.

## 1 Introduction

Normal-form games involve multiple agents who independently choose their actions from a finite set of options, while their rewards depend on the joint actions of all agents. Thus, when they learn their strategies, it matters to take into account the temporal change in the others' strategies. One of the representative methods in such multi-agent learning is "optimistic" algorithms, where an agent not only updates its strategy naively by the current rewards, such as Follow the Regularized Leader (FTRL) [1, 2] and Mirror Descent (MD) [3, 4], but also predicts its future reward. These algorithms are called optimistic FTRL (OFTRL) [5] and MD (OMD) [6] and are known to achieve good performance in two metrics. The first metric is regret, which evaluates how close to the optimum the time series of their strategies is. When all agents adopt OFTRL or OMD, their social regret is constant regardless of the final time $T$ [6, 5], which outperforms the regret of $O(\sqrt{T})$ by vanilla FTRL and MD [7, 8]. The second metric is convergence, which judges whether or not their strategies converge to the Nash equilibrium. OFTRL and OMD are also known to converge to the Nash equilibrium in poly-matrix zero-sum games [9, 10, 11, 12], whereas vanilla FTRL and MD frequently exhibit cycling and fail to converge [13, 14, 15]. Therefore, the prediction of future reward plays a key role in multi-agent learning.

In such optimistic algorithms, the prediction of future reward is based on observed previous rewards. In the real world, however, various factors can hinder the observation of previous rewards. One of the

most likely causes of this unobservability is time delay (or latency) to observe the previous rewards. A typical situation where such time delays in learning in games matter is a market (i.e., the Cournot competition [16, 17, 18]), where each firm determines the amount of its products as its strategy, but sales of the products can be observed with some time lags. In addition, multi-agent recommender systems are often used and analyzed [19, 20], where multiple recommenders equipped with different perspectives cooperatively introduce an item to buyers. Such recommender systems also face the problem of time-delayed feedback [21] because some delays occur until the buyers actually purchase the item. Such delayed feedback perhaps makes it difficult to predict future rewards and worsens the performance of multi-agent learning.

This study proposes and addresses the problem that time-delayed feedback significantly impacts both social regret and convergence in learning in games. Our contributions are summarized as follows.

- **An example is provided where any small time delay worsens both social regret and convergence.** This example is Matching Pennies under the unconstrained setting with the Euclidean regularizer. In detail, Thm. 6 shows that OFTRL suffers from $O(\sqrt{T})$-regret, corresponding to the worst-case regret obtained in the adversarial setting. In addition, Thm. 7 shows that OFTRL cannot converge to the equilibrium but rather diverges.

- **An algorithm tolerant to time delay is proposed and interpreted.** We formulate weighted OFTRL (WOFTRL), which weights the optimism in OFTRL by $n$ times. Based on the Taylor expansion, we find that the time delay $m$ and the optimistic weight $n$ cancel each other out. This cancel-out is also confirmed in Thms. 6 and 7.

- **Our algorithm is proved to achieve the constant social regret and last-iterate convergence.** In Cor. 11, we prove that when the optimistic weight one-step exceeds the time delay $(n = m + 1)$, the regret is $O(m^2)$. Furthermore, Cor. 13 shows that the strategies converge to the Nash equilibrium in any poly-matrix zero-sum games. Our experiments (Figs. 1-3) also support and reinforce these corollaries.

**Related works:** Delayed feedback has been frequently studied in the context of online learning. The delays are known to worsen regret in the full feedback [22, 23, 24, 25, 26] and bandit feedback [27, 28, 29, 30, 31, 32, 33, 34] settings. In the context of learning in games, abnormal feedback has recently attracted attention and been studied. A representative example is time-varying games, where a gap between the true reward and feedback always exists as the game changes over time. Such a gap affects the properties of regret [35, 36, 37, 38] and convergence [39, 40, 41, 42].

## 2 Setting

We consider $N$ players which is labeled by $i \in \{1, \cdots, N\}$. Let $\mathcal{A}_i$ denote the set of player $i$'s actions. Each player $i$'s payoff depends on its own action and the actions of the other players. Each player $i$'s strategy is in what probabilities the player chooses its actions and given by the probability distribution $\boldsymbol{x}_i \in \mathcal{X}_i := \Delta^{|\mathcal{A}_i|-1}$. The expected payoff is defined as $U_i(\boldsymbol{x}_1, \cdots, \boldsymbol{x}_N)$. Here, $U_i$ is multi-linear; in other words, the following equation holds

$$U_i(\boldsymbol{x}_1, \cdots, a\boldsymbol{x}_j + a'\boldsymbol{x}'_j, \cdots, \boldsymbol{x}_N) = aU_i(\boldsymbol{x}_1, \cdots, \boldsymbol{x}_j, \cdots, \boldsymbol{x}_N) + a'U_i(\boldsymbol{x}_1, \cdots, \boldsymbol{x}'_j, \cdots, \boldsymbol{x}_N),$$
(1)

for all $\boldsymbol{x}_j, \boldsymbol{x}'_j \in \mathcal{X}_i$ and all $a, a' \in \mathbb{R}$ and all $j \in \{1, \cdots, N\}$.

We define the gradient of the expected payoff as $\boldsymbol{u}_i(\boldsymbol{x}_1, \cdots, \boldsymbol{x}_N) = \partial U_i(\boldsymbol{x}_1, \cdots, \boldsymbol{x}_N)/\partial \boldsymbol{x}_i$. (Here, we remark that $\boldsymbol{u}_i(\boldsymbol{x}_1, \cdots, \boldsymbol{x}_N)$ is independent of $\boldsymbol{x}_i$ because of the multi-linearity of $U_i$.) We use the concatenation of the strategies and rewards as $\boldsymbol{x} := (\boldsymbol{x}_1, \cdots, \boldsymbol{x}_N) \in \mathcal{X} := (\mathcal{X}_1, \cdots, \mathcal{X}_N)$ and $\boldsymbol{u}(\boldsymbol{x}) = (\boldsymbol{u}_1(\boldsymbol{x}), \cdots, \boldsymbol{u}_N(\boldsymbol{x}))$, respectively. Based on the multi-linearity of $U_i$, the $L$-Lipschitz continuity is satisfied with $L := 2\max_{(i,\boldsymbol{x})}|U_i(\boldsymbol{x})|$ as

$$\|\boldsymbol{u}(\boldsymbol{x}) - \boldsymbol{u}(\boldsymbol{x}')\|_2 \leq L\|\boldsymbol{x} - \boldsymbol{x}'\|_2.$$
(2)

### 2.1 Online Learning with Time Delay

Every round $t \in \{1, \cdots, T\}$, each player sequentially determines its strategy $\boldsymbol{x}_i^t \in \mathcal{X}_i$. As a result, the player observes the rewards for each action, i.e., $\boldsymbol{u}_i^t = \boldsymbol{u}_i(\boldsymbol{x}_1^t, \cdots, \boldsymbol{x}_N^t)$ (called full feedback

setting). Here, $u_{ij}^t$ indicates player $i$'s reward by choosing action $a_j$. Thus, in online learning, each player determines the next strategy $\boldsymbol{x}_i^{t+1}$ by using the algorithm $\boldsymbol{f}_i$ of the past rewards $\{\boldsymbol{u}_i^s\}_{1 \le s \le t}$, which is denoted as

$$\boldsymbol{x}_i^{t+1} = \boldsymbol{f}_i(\{\boldsymbol{u}_i^s\}_{1 \le s \le t}). \tag{3}$$

Let us further define a time delay in the observation of past rewards. When the time delay of $m \in \mathbb{N}$ steps occurs, only the rewards of $\{\boldsymbol{u}_i^{s-m}\}_{1 \le s \le t}$ are observable in Eq. (3). Here, we assumed that rewards before the initial time give no information, i.e., $\boldsymbol{u}_i^t = \boldsymbol{0}$ for all $t \le 0$. Thus, the time delay modifies online learning algorithms as follows.

**Definition 1** (Online learning with time delay). *With the time delay of $m \in \mathbb{N}$ steps, online learning is modified as*

$$\boldsymbol{x}_i^{t+1} = \boldsymbol{f}_i(\{\boldsymbol{u}_i^{s-m}\}_{1 \le s \le t}), \tag{4}$$

*where $\boldsymbol{u}_i^t = \boldsymbol{0}$ for all $t \le 0$.*

## 2.2 Follow the Regularized Leader

Under the full feedback setting, one of the most successful algorithms is Follow the Regularized Leader (FTRL). The generalization of this FTRL is formulated as follows.

**Definition 2** (Generalized FTRL with time delay). *With the time delay of $m \in \mathbb{N}$ steps, generalized FTRL [5] is formulated as follows*

$$\boldsymbol{x}_i^{t+1} = \arg \max_{\boldsymbol{x}_i \in \mathcal{X}} \eta \left\langle \boldsymbol{x}_i, \sum_{s=1}^{t-m} \boldsymbol{u}_i^s + \boldsymbol{m}_i^t \right\rangle - h(\boldsymbol{x}_i). \tag{5}$$

*Here, $h(\boldsymbol{x}_i)$ is 1-strongly convex and is called regularizer.*

Here, $\eta \in \mathbb{R}$ is learning rate, and $h$ is a regularizer. $\boldsymbol{m}_i^t$ depends on the details of FTRL variants, typically, vanilla [1, 2] and optimistic FTRL [5].

**Algorithm 3** (Vanilla FTRL with time delay). *When $\boldsymbol{m}_i^t = \boldsymbol{0}$, generalized FTRL corresponds to vanilla FTRL.*

**Algorithm 4** (Optimistic FTRL with time delay). *When $\boldsymbol{m}_i^t = \boldsymbol{u}_i^{t-m}$, generalized FTRL corresponds to optimistic FTRL (OFTRL).*

# 3 Issue by Time Delay

In this section, we introduce an example in which any small time delay worsens the performance of OFTRL. As an example, we define Matching Pennies as follows. This is the simplest game that provides cycling behavior by vanilla FTRL.

**Example 5** (Matching Pennies). *Matching Pennies considers two players ($N = 2$) with their rewards;*

$$\boldsymbol{u}_1 = +\boldsymbol{A}\boldsymbol{x}_2, \quad \boldsymbol{u}_2 = -\boldsymbol{A}^{\mathrm{T}}\boldsymbol{x}_1, \quad \boldsymbol{A} = \begin{pmatrix} +1 & -1 \\ -1 & +1 \end{pmatrix}. \tag{6}$$

Let us focus on two aspects: regret and convergence. First, individual ($\mathrm{REG}_i$) and social ($\mathrm{REGTOT}$) regret are defined as

$$\mathrm{REG}_i(T) := \max_{\boldsymbol{x}_i \in \mathcal{X}_i} \sum_{t=1}^{T} \left\langle \boldsymbol{x}_i - \boldsymbol{x}_i^t, \boldsymbol{u}_i^t \right\rangle, \quad \mathrm{REGTOT}(T) := \sum_{i=1}^{N} \mathrm{REG}_i(T). \tag{7}$$

It has already been known that without any time delay, OFTRL achieves constant social regret (called "fast convergence" [5]). However, if a slight time delay exists, OFTRL suffers from the social regret of $\Omega(\sqrt{T})$ at least even in such a simple game (see Appendix B for the full proof). This regret is also obtained under an adversarial environment [5] and thus is worst-case for OFTRL.

**Theorem 6** (Social regret of $\Omega(\sqrt{T})$ by time delay). *Suppose that the strategy space is unconstrained ($\boldsymbol{x}_i \in \mathbb{R}^{|\mathcal{A}_i|}$) in Exm. 5 with the Euclidean regularizer ($h(\boldsymbol{x}_i) = \|\boldsymbol{x}_i\|_2^2/2$). Then, when both players use OFTRL ($n = 1$) with any time delay ($m \geq 1$), their social regret is no less than $\Omega(\sqrt{T})$, which is achieved with $\eta = 1/\sqrt{T}$.*

*Proof Sketch.* In the proof, we consider an extended class of OFTRL, where the weight of the optimistic prediction of the future is generalized. By direct calculations, we obtain the recurrence formula of the dynamics of both players' strategies. We prove a lemma that this recurrence formula is approximately solved by circular functions, whose radius varies with time depending on the time delay $m$ and optimistic weight $n$. Here, we emphasize that the rate of the radius change is $e^{\alpha t}$ with $\alpha = m - n + 1/2$. Thus, if the players use OFTRL ($n = 1$), the radius always grows ($\alpha > 0$) for any time delay $m \geq 1$. In conclusion, OFTRL performs the same as vanilla FTRL and suffers from the social regret of $\Omega(\sqrt{T})$ at least. $\qquad\square$

Second, the Nash equilibrium is defined as $\boldsymbol{x}^* = (\boldsymbol{x}_1^*, \cdots, \boldsymbol{x}_N^*)$ which satisfies

$$\boldsymbol{x}_i^* \in \arg \max_{\boldsymbol{x}_i \in \mathcal{X}_i} U_i(\boldsymbol{x}_1^*, \cdots, \boldsymbol{x}_i, \cdots, \boldsymbol{x}_N^*), \tag{8}$$

for all $i = 1, \cdots, N$. We also denote the set of Nash equilibria as $\mathcal{X}^*$. In poly-matrix zero-sum games, where the payoff $U_i$ is divided into the zero-sum matrix games, it has been known that OFTRL achieves convergence to the Nash equilibrium. This convergence is measured by the distance from the Nash equilibria, which is formulated as

$$\text{DIS}(T) := \sqrt{\min_{\boldsymbol{x}^* \in \mathcal{X}^*} \sum_{i=1}^N \|\boldsymbol{x}_i^T - \boldsymbol{x}_i^*\|_2^2}. \tag{9}$$

Under the existence of any small time delay ($m \geq 1$), the distance ($\text{DIS}(T)$) does not converge but rather diverges as follows (see Appendix C for the full proof).

**Theorem 7** (Divergence by time delay). *Suppose that the strategy space is unconstrained ($\boldsymbol{x}_i \in \mathbb{R}^{|\mathcal{A}_i|}$) in Exm. 5 with the Euclidean regularizer ($h(\boldsymbol{x}_i) = \|\boldsymbol{x}_i\|_2^2/2$). Then, when both the players use OFTRL ($n = 1$) with any time delay ($m \geq 1$), their strategies diverge from the equilibrium ($\lim_{T \to \infty} \text{DIS}(T) \to \infty$).*

*Proof Sketch.* We reuse the lemma in the proof of Thm. 6, showing that the dynamics of the strategies of both players are approximately solved by circular functions, whose radius varies with the exponential rate of $\alpha = m - n + 1/2$. In OFTRL ($n = 1$) with some delay ($m \geq 1$), the rate is positive ($\alpha > 0$), meaning that the radius is amplified with time. Because the equilibrium is at the center of the circular function, the divergence from the equilibrium is proven. $\qquad\square$

**Interpretation of Thms. 6 and 7:** The crux of their proofs is that the exponential rate of the radius change ($\alpha$) is divided into three terms: 1) expansion by the time delay ($+m$), 2) contraction by optimism ($-n$), and 3) expansion by the accumulation of the discretization errors. These terms intuitively explain the previous and present results. First, in the case of vanilla FTRL ($m = n = 0$), the radius grows with the exponential rate of $\alpha = 1/2$. This means that the strategies diverge from the Nash equilibrium over time, and social regret converges only slowly. Second, in the case of OFTRL ($m = 0$, $n = 1$), the radius shrinks with the exponential rate of $\alpha = -1/2$. This means that the strategies converge to the equilibrium over time, and social regret becomes constant. Finally, if OFTRL is accompanied by a time delay ($m \geq 1$, $n = 1$), the radius grows with the exponential rate of $\alpha \geq 1/2$, resulting in the social regret of $\Omega(\sqrt{T})$ again.

**Remark on unconstrained setting:** Note that Thm. 6 considers the unconstrained setting. In other words, the dynamics assume no boundary condition in the strategy space. This unconstrained setting is necessary to analyze the global behavior of the dynamics, which differs on the boundary of the strategy space in the constrained setting. However, this unconstrained setting is sufficient to capture the worsening of the performance of OFTRL. At least, our experiments observe the same results as Thms. 6 and 7 even in the constrained setting.

# 4 Algorithm

So far, we understand an issue that OFTRL becomes useless due to the effect of time delay. The next question is how the property of constant social regret is recovered, even under time delay. In the following, we propose an extension of OFTRL, where the optimistic prediction of the future reward is added $n \in \mathbb{N}$ times.

**Algorithm 8** (Weighted Optimistic Follow the Regularized Leader). *Weighted Optimistic Follow The Regularized Leader (WOFTRL) is given by $\boldsymbol{m}_i^t = n\boldsymbol{u}_i^{t-m}$ for $n \in \mathbb{N}$ in generalized FTRL.*

**Interpretation of WOFTRL:** WOFTRL can be rewritten as

$$\boldsymbol{x}_i^t = \arg \max_{\boldsymbol{x}_i \in \mathcal{X}_i} \eta \left\langle \boldsymbol{x}_i, \tilde{\boldsymbol{x}}_i^t \right\rangle - h(\boldsymbol{x}_i), \quad \tilde{\boldsymbol{x}}_i^t = \sum_{s=1}^{t-m-1} \boldsymbol{u}_i^s + n\boldsymbol{u}_i^{t-m-1}. \tag{10}$$

For arbitrary time series $\{v^t\}_t$, we define the finite difference of the time series as

$$\delta v^t := v^{t+1} - v^t, \quad \delta\delta v^t = \delta v^{t+1} - \delta v^t. \tag{11}$$

In Eq. (10), the following relation is satisfied;

$$\delta\tilde{\boldsymbol{x}}_i^t = \boldsymbol{u}_i^{t-m} + n\delta\boldsymbol{u}_i^{t-m-1}. \tag{12}$$

By definition, we evaluate the time-delayed terms as

$$\boldsymbol{u}_i^{t-m} = \boldsymbol{u}_i^t - m\delta\boldsymbol{u}_i^t - \sum_{s=t-m}^{t-1}\sum_{r=s}^{t-1} \delta\delta\boldsymbol{u}_i^r, \quad \delta\boldsymbol{u}_i^{t-m-1} = \delta\boldsymbol{u}_i^t - \sum_{s=t-m-1}^{t-1} \delta\delta\boldsymbol{u}_i^s. \tag{13}$$

By substituting these into Eq. (12), we obtain

$$\delta\tilde{\boldsymbol{x}}_i^t = \boldsymbol{u}_i^t + (n-m)\delta\boldsymbol{u}_i^t - \left( \sum_{s=t-m}^{t-1}\sum_{r=s}^{t-1} \delta\delta\boldsymbol{u}_i^r + n\sum_{s=t-m-1}^{t-1} \delta\delta\boldsymbol{u}_i^s \right). \tag{14}$$

In the RHS, the first term corresponds to the vanilla FTRL without time delay and is $O(1)$. Here, it is known that predicting future rewards enhances performance in terms of social regret and convergence. Since $\delta\boldsymbol{u}_i^t$ in the second term shows the time derivative of rewards and is $O(\eta)$, it pushes ahead the time $t$ in $\boldsymbol{u}_i^t$. Thus, if its coefficient is positive ($n - m > 0$), the second term contributes to predicting a future reward and is expected to improve performance. In addition, we remark that an optimistic weight $n$ and time delay $m$ conflict with each other. Finally, the third term is a prediction error which consists of the accumulation of only $\delta\delta\boldsymbol{u}_i^t$ and is $O(\eta^2)$. Thus, this term is negligible if $\eta$ is sufficiently small.

# 5 Constant Social Regret

This section shows that our WOFTRL can achieve constant social regret. For the regret analysis, let us introduce the following Regret bounded by Variation in Utilities (RVU) property [5].

**Definition 9** (RVU property). *The time series of $\{\boldsymbol{x}_i^t\}_{1 \le t \le T}$ is defined to satisfy the RVU property if there exists $0 < \alpha$ and $0 < \beta \le \gamma$ such that*

$$\mathrm{REG}_i(T) \le \alpha + \beta \sum_{t=1}^{T} \|\boldsymbol{u}_i^t - \boldsymbol{u}_i^{t-1}\|_2^2 - \gamma \sum_{t=1}^{T} \|\boldsymbol{x}_i^t - \boldsymbol{x}_i^{t-1}\|_2^2. \tag{15}$$

We can prove that WOFTRL satisfies this property as follows (see Appendix D for the full proof).

**Theorem 10** (RVU property when $n = m + 1$). *When $n = m + 1$, WOFTRL using learning rate $\eta$ satisfies the RVU property with constraints $\alpha = h_{\max}/\eta$, $\beta = \lambda^2\eta$, and $\gamma = 1/(4\eta)$. Here, we defined $\lambda := (m+1)(m+2)/2$.*

*Proof sketch.* The basic procedure of the proof is the same as the previous study [5]. We show that when $n = m + 1$, the cumulative payoff ($\tilde{\boldsymbol{x}}_i^t$ in Eq. (10)) well approximates the gold reward until time $t$, defined as

$$\tilde{\boldsymbol{g}}_i^t := \sum_{s=1}^{t} \boldsymbol{u}_i^s. \tag{16}$$

Here, note that $\beta$ is $\lambda^2$ times larger (worse) than without the time delay. $\lambda$ is interpreted as accumulated errors in predicting the gold reward. Indeed, the difference between the gold and predicted rewards is described as

$$\tilde{\boldsymbol{g}}_i^t - \tilde{\boldsymbol{x}}_i^t = \sum_{s=t-m}^{t} \boldsymbol{u}_i^s - (m+1)\boldsymbol{u}_i^{t-m-1}, \tag{17}$$

where the total time gap in its RHS is evaluated as $\sum_{s=t-m}^{t}\{s-(t-m-1)\} = (m+1)(m+2)/2 = \lambda$. □

**Corollary 11** (Constant social regret with time delay). *When $n = m+1$, WOFTRL using the learning rate of $\eta = 1/(2\lambda L)$ achieves the social regret of $O(m^2)$, i.e., $\text{REGTOT}(T) = O(m^2)$.*

*Proof.* Sum up for $i \in \{1, \cdots, N\}$ the RVU property in Thm. 10, then the second term of the payoff is lower-bounded by the $L$-Lipschitz continuity as

$$\|\boldsymbol{u}^t - \boldsymbol{u}^{t-1}\|_2^2 \leq L^2 \|\boldsymbol{x}^t - \boldsymbol{x}^{t-1}\|_2^2. \tag{18}$$

By directly applying $\eta = 1/(2\lambda L)$, the second and third terms are canceled out, and we obtain

$$\text{REGTOT}(T) \leq 2\lambda N L h_{\max} = O(m^2). \tag{19}$$

□

**Remark on Cor. 11:** An important remark is that the time delay $m$ worsens social regret. One demerit is that we must use a $\lambda = (m+1)(m+2)/2$ times smaller learning rate. This leads to another demerit that the social regret becomes $\lambda = (m+1)(m+2)/2$ times larger, too. In addition, we remark that if no time delay exists ($m = 0$), the previous result (Cor. 6 in [5]) is restored as $\lambda = 1$. We also obtain the individual regret of $O(m^{3/2}T^{1/4})$ by utilizing Thm. 10 (see Appendix F for details).

## 5.1 Experiment for Matching Pennies

We consider Matching Pennies defined in Exm. 5. Now Fig. 1 provides the experiments for WOFTRL with the Euclidean regularizer, i.e., $h(\boldsymbol{x}_i) = \|\boldsymbol{x}_i\|_2^2/2$.

**Constant social regret when $n > m$:** First, see the upper left triangle region of Panel A. This region corresponds to $n > m$, where the optimistic weight is greater than the time delay. It shows that the regret is sufficiently small. In detail, Panel B shows the time series of the regrets for various optimistic weights $n$ with a fixed time delay ($m = 10$). In the cases of $n > m$ (i.e., $n = 11, 13, 15$), we see that the regrets converge to sufficiently small values. Finally, Panel C shows the two ways to take the learning rate $\eta$. When we keep $\eta$ constant, the regret is also constant ($O(1)$), independent of the final time $T$. This is completely different from the case of $\eta = 1/\sqrt{T}$, where the regret grows with $O(\sqrt{T})$.

$O(\sqrt{T})$**-social regret when $n \leq m$:** Next, see the lower right triangle region of Panel A. This region corresponds to $n \leq m$, where the optimistic weight is smaller than the time delay. It shows that the regret takes various values and is relatively large. In detail, Panel B shows that the regret oscillates and is large in the case of $n \leq m$ (i.e., $n = 1, 3, 5, 7, 9$ against $m = 10$). It also shows that a transition in the behavior of the regret occurs at $n = m$.

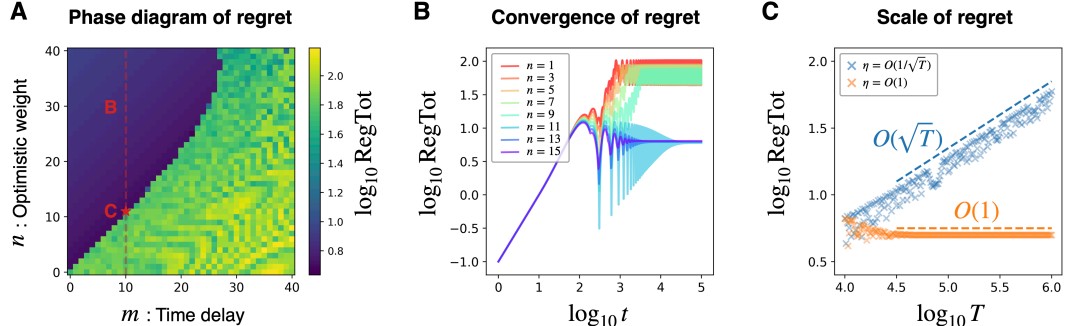

Figure 1: Regret analysis for Matching Pennies (Exm. 5). **A**. The phase diagram of social regrets for various time delays $m$ (horizontal) and optimistic weights $n$ (vertical). The deep blue color indicates that the regret is small ($O(1)$-regret), while the green and yellow ones indicate that the regret is large ($O(\sqrt{T})$-regret). A transition is clearly shown between $O(1)$- and $O(\sqrt{T})$-regret. We set the parameters as $T = 10^5$ and $\eta = 10^{-2}$. **B**. The convergence of social regrets for various optimistic weights $n$ and a fixed time-delay $m = 10$ (corresponding to the red broken line in Panel A). We see the regret oscillates and is relatively large for $n = 1, 3, 5, 7, 9$ ($O(\sqrt{T})$-regret) but converges to a small value for $n = 11, 13, 15$ ($O(1)$-regret). A transition is clearly observed again in $m = n$. We set the parameters as $\eta = 10^{-2}$. **C**. The scale of social regrets in the case of $m = 10$ and $n = 11$ (corresponding to the red star in Panel A). We plot the two ways to take learning rate: $\eta = 1/\sqrt{T}$ (blue dots) and $\eta = O(1)$ (orange ones). The regrets for $\eta = O(1/\sqrt{T})$ follow the broken blue line, which has a slope of $1/2$ (meaning that the regrets are $O(\sqrt{T})$). On the other hand, the regrets for $\eta = O(1)$ follow the broken orange line, which has a slope of $0$ (meaning that the regrets are $O(1)$). We set $\eta = 1/\sqrt{T}$ for the blue dots and $\eta = 10^{-2}$ for the orange ones.

**Exceptional $O(\sqrt{T})$-social regret when $n > m$:** Finally, see the particular region where $m$ is large but $n - m$ is sufficiently small (e.g., $m = 30$ and $n = 35$) in Panel A. Note that the regret is $O(\sqrt{T})$ even though this region satisfies $n > m$. This $O(\sqrt{T})$-regret is due to the finiteness of the learning rate $\eta$. When there is a sufficiently large time delay (e.g., $m = 30$) against a finite learning rate (e.g., $\eta = 10^{-2}$ in Panel A), it becomes difficult to estimate the current status from the time-delayed rewards. This difficulty is related to the accumulation of estimation errors following $\lambda = (m + 1)(m + 2)/2$ in Cor. 11. If we take sufficiently small $\eta$, the fast convergence occurs in the wider region of $n > m$.

### 5.2 Experiment for Sato's Game

We further discuss an example of a nonzero-sum game. We pick up a Sato's game [43], which is Rock-Paper-Scissors, but some scores are also generated in draw cases, as follows.

$$\boldsymbol{u}_1 = +(\boldsymbol{A} + \boldsymbol{D})\boldsymbol{x}_2, \quad \boldsymbol{u}_2 = -\boldsymbol{A}^{\mathrm{T}}\boldsymbol{x}_1, \quad \boldsymbol{A} = \begin{pmatrix} 0 & -1 & +1 \\ +1 & 0 & -2 \\ -1 & +2 & 0 \end{pmatrix}, \quad \boldsymbol{D} = \frac{1}{10}\begin{pmatrix} 1 & 0 & 0 \\ 0 & 2 & 0 \\ 0 & 0 & 2 \end{pmatrix}. \tag{20}$$

In such nonzero-sum Sato's games, heteroclinic cycles with complex oscillations that diverge from the Nash equilibrium are observed [43].

From our experiments (Fig. 2), we obtain similar results to those in Fig. 1: (i) Constant social regret when $n > m$, (ii) $O(\sqrt{T})$-social regret when $n \leq m$, and (iii) Exceptional $O(\sqrt{T})$-social regret when $n > m$.

## 6 Convergence in Poly-Matrix Zero-Sum Games

We also show that WOFTRL convergences to the Nash equilibrium in poly-matrix zero-sum games, which are separable into zero-sum games between a couple of players, as defined below.

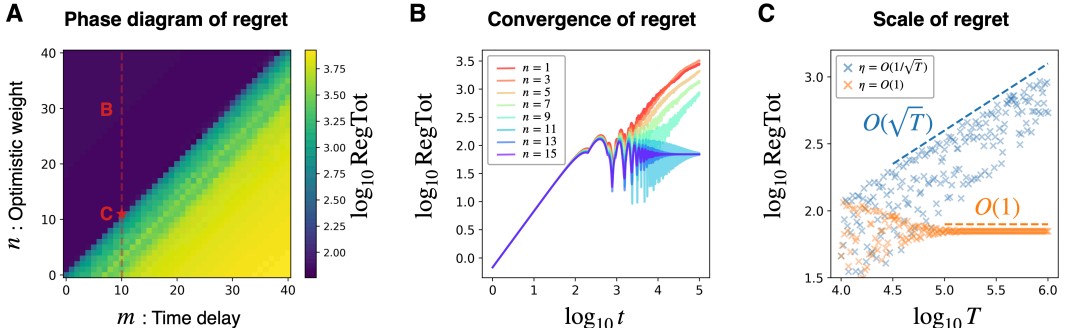

Figure 2: Regret analysis for a Sato's game (Eqs. 20) with the entropic regularizer, i.e., $h(\boldsymbol{x}) = \langle \boldsymbol{x}, \log \boldsymbol{x} \rangle$. The results and parameter settings for all the panels are the same as those in Fig. 1. **A**. The phase diagram of social regrets for various time delays $m$ (horizontal) and optimistic weights $n$ (vertical). **B**. The convergence of social regrets for various optimistic weights $n$ and a fixed time-delay $m = 10$ (corresponding to the red broken line in Panel A). **C**. The scale of social regrets in the case of $m = 10$ and $n = 11$ (corresponding to the red star in Panel A).

**Definition 12** (Poly-matrix zero-sum games). *Poly-matrix zero-sum games are*

$$\boldsymbol{u}_i = \sum_{i' \neq i} \boldsymbol{A}_{(ii')} \boldsymbol{x}_{i'}, \quad \boldsymbol{A}_{(i'i)} = -\boldsymbol{A}_{(ii')}^{\mathrm{T}}. \tag{21}$$

Matching Pennies is a special case of poly-matrix games. In poly-matrix zero-sum games, the strategy converges to the Nash equilibrium for a sufficiently small learning rate (see Appendix E for the full proof).

**Corollary 13** (Last-Iterate Convergence). *Suppose $h$ is a convex function of Legendre type [44, 45]. When $n = m + 1$, WOFTRL using the learning rate of $\eta \leq 1/(\sqrt{8}n^2 L)$ converges to the Nash equilibrium, i.e., there exists $\boldsymbol{x}^* \in \mathcal{X}^*$ such that $\lim_{T \to \infty} \|\boldsymbol{x}^T - \boldsymbol{x}^*\|_2 = 0$.*

*Proof Sketch.* We follow but extend the method by the prior study [10]. First, we prove that when the regularizer is Legendre type, generalized FTRL is equivalent to its MD (mirror descent) variant [6]. (We remark that this is an extension of the already-known equivalence between vanilla FTRL and MD [46, 45].) We thereafter show the last-iterate convergence via this MD variant. Furthermore, for each Nash equilibrium $\boldsymbol{x}^* \in \mathcal{X}^*$, we find a divergence $V^t(\boldsymbol{x}^*)$ defined as

$$V^t(\boldsymbol{x}^*) := D(\boldsymbol{x}^*, \hat{\boldsymbol{x}}^t) + \frac{1}{2} \Big\{ D(\hat{\boldsymbol{x}}^t, \boldsymbol{x}^{t-1}) + \sum_{r=1}^{n-1} \frac{n-r}{n} (D(\boldsymbol{x}^{t-r}, \hat{\boldsymbol{x}}^{t-r}) + D(\hat{\boldsymbol{x}}^{t-r}, \boldsymbol{x}^{t-r-1})) \Big\}, \tag{22}$$

which monotonically decreases with time $t$. We prove that this divergence decreases to $0$ for one equilibrium $\boldsymbol{x}^*$, meaning that WOFTRL converges to the equilibrium. $\square$

### 6.1 Experiment for Rock-Paper-Scissors

Now, Fig. 3 shows the simulations for weighted Rock-Paper-Scissors, formulated as Eqs. (20) with the nonzero-sum term $\boldsymbol{D}$ eliminated.

**Last-iterate convergence when $n > m$:** See the two right panels, where the optimistic weight ($n = 5, 6$) dominates the time delay ($m = 4$). Thus, the trajectories converge to the Nash equilibrium, meaning last-iterate convergence. Here, we also observe that when the optimistic weight is larger ($n = 6$ than $n = 5$), convergence is faster.

**Non-convergence when $n \leq m$:** See the two left panels, where the optimistic weight ($n = 3, 4$) does not dominate the time delay $m = 4$. Thus, the trajectories fail to converge but rather diverge from the equilibrium. Here, we also see that when the optimistic weight is smaller ($n = 3$ than $n = 4$), the divergence is faster.

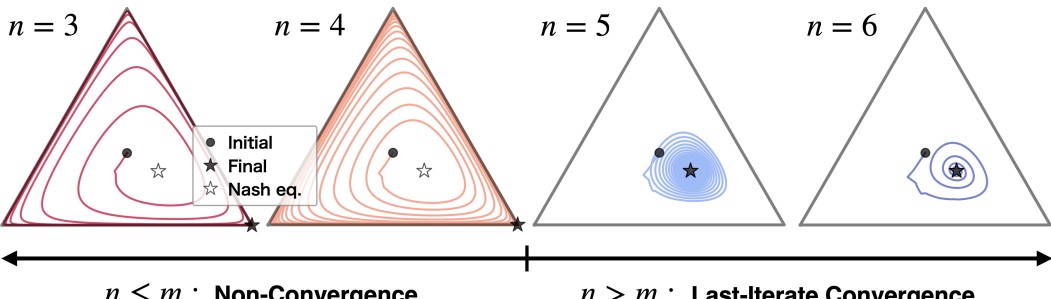

Figure 3: Convergence analysis for a weighted Rock-Paper-Scissors with the entropic regularizer, i.e., $h(\boldsymbol{x}_i) = \langle \boldsymbol{x}_i, \log \boldsymbol{x}_i \rangle$. We set the parameters as $\eta = 10^{-1}$ and $m = 4$. The colored lines are the trajectories of learning. The black dot, black star, and white star indicate the initial state, final state, and Nash equilibrium, respectively. From left to right, optimistic weights are $n = 3, 4, 5, 6$. In the left two panels ($n \leq m$), the black star does not overlap the white one, meaning non-convergence. On the other hand, in the right two panels ($n > m$), the black star overlaps the white one, indicating last-iterate convergence.

## 7 Conclusion

This study tackled the problem of time-delayed full feedback in learning in games. First, we demonstrated, in a simple example of Matching Pennies with the unconstrained setting, that any slight time delay makes the performance of OFTRL worse from the perspectives of social regret (Thm. 6) and convergence (Thm. 7). To overcome these impossibility theorems, we introduced WOFTRL, which weights the optimism of OFTRL $n$ times. The Taylor expansion for the learning rate revealed that the optimistic weight $n$ cancels the time delay $m$ out (Eq. (14)). We proved that when the optimistic weight even slightly dominates the time delay ($n = m + 1$), both constant social regret (Cor. 11) and last-iterate convergence (Cor. 13) hold. Our experiments (Figs. 1-3) strengthened the results of these corollaries.

One future direction is to consider other types of time delay. Although this study only considered that the delay $m$ is a constant value, real-world delays are often more complicated. For example, previous studies on online learning have discussed situations where $m$ is given stochastically [31, 32] or adversarially [24, 25, 26]. We remark that the theory of this study is to some extent applicable to such stochastic delays; It is sufficient to take a larger $n$ than the possible delays and use the reward before $n - 1$ steps every time, i.e., use GFTRL with $\boldsymbol{m}_i^t = n\boldsymbol{u}_i^{t-n+1}$. Another direction is to obtain the convergence rate for WOFTRL. This study opens the door to time-delay problems and offers various straightforward open questions in the field of algorithmic game theory.

## Acknowledgments and Disclosure of Funding

Kaito Ariu was supported by JSPS KAKENHI Grant Numbers 23K19986 and 25K21291.

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

# Appendix

## A Dynamics in Unconstrained, Euclidean, Matching Pennies

Suppose the unconstrained setting for the Euclidean regularizer $h(\boldsymbol{x}_i) = \|\boldsymbol{x}_i\|_2^2/2$, then the convex-conjugate

$$\boldsymbol{x}_i^t = \arg\max_{\boldsymbol{x}_i \in \mathbb{R}^d} \eta \left\langle \boldsymbol{x}_i, \tilde{\boldsymbol{x}}_i^t \right\rangle - h(\boldsymbol{x}_i), \tag{A1}$$

is trivially solved as

$$\boldsymbol{x}_i^t = \tilde{\boldsymbol{x}}_i^t. \tag{A2}$$

We further consider Matching Pennies, where the payoff matrix is

$$\boldsymbol{A} = \begin{pmatrix} +1 & -1 \\ -1 & +1 \end{pmatrix} = \mathbf{c} \otimes \mathbf{c}, \quad \mathbf{c} := \begin{pmatrix} +1 \\ -1 \end{pmatrix}. \tag{A3}$$

By substituting Eq. (A2) into Eq. (12), the dynamics of the strategies are described as

$$\boldsymbol{x}_i^{t+1} = \boldsymbol{x}_i^t + \eta \boldsymbol{u}_i^{t-m} + n\eta(\boldsymbol{u}_i^{t-m} - \boldsymbol{u}_i^{t-m-1}), \tag{A4}$$

To simplify Eq. (A4), let us define

$$\Delta x_i^t := \left\langle \boldsymbol{x}_i^t, \mathbf{c} \right\rangle. \tag{A5}$$

Then, $\boldsymbol{u}_i^t$ is calculated as

$$\boldsymbol{u}_1^t = +\boldsymbol{A}\boldsymbol{x}_2^t = +\Delta x_2^t \mathbf{c}, \quad \boldsymbol{u}_2^t = -\boldsymbol{A}^{\mathrm{T}}\boldsymbol{x}_1^t = -\Delta x_1^t \mathbf{c}. \tag{A6}$$

Finally, the following recurrence formulas are obtained;

$$\Delta x_1^{t+1} = \Delta x_1^t + 2(n+1)\eta \Delta x_2^{t-m} - 2n\eta \Delta x_2^{t-m-1}, \tag{A7}$$

$$\Delta x_2^{t+1} = \Delta x_2^t - 2(n+1)\eta \Delta x_1^{t-m} + 2n\eta \Delta x_1^{t-m-1}. \tag{A8}$$

These recurrence formulas are approximately solved as follows.

**Lemma A1** (Solution of recurrence formulas). *Eqs. (A7) and (A8) are solved as*

$$\Delta x_1^t = e^{\lambda_t}\cos\theta_t, \quad \Delta x_2^t = -e^{\lambda_t}\sin\theta_t, \tag{A9}$$

$$\lambda_{t+1} = \lambda_t + \left(m - n + \frac{1}{2}\right)(2\eta)^2 + \Theta(\eta^4), \quad \theta_{t+1} = \theta_t + 2\eta + \Theta(\eta^3), \tag{A10}$$

*Proof.* For readability, we scale the learning rate in Eqs. (A7) and (A8) as $2\eta \leftarrow \eta$, we obtain

$$\Delta x_1^{t+1} = \Delta x_1^t + (n+1)\eta \Delta x_2^{t-m} - n\eta \Delta x_2^{t-m-1}, \tag{A11}$$

$$\Delta x_2^{t+1} = \Delta x_2^t - (n+1)\eta \Delta x_1^{t-m} + n\eta \Delta x_1^{t-m-1}. \tag{A12}$$

By using the relation of Eqs. (A9), we obtain

$$e^{\lambda_{t+1}+i\theta_{t+1}} = \Delta x_1^{t+1} - i\Delta x_2^{t+1} \tag{A13}$$

$$= \left\{\Delta x_1^t + (n+1)\eta \Delta x_2^{t-m} - n\eta \Delta x_2^{t-m-1}\right\}$$
$$- i\left\{\Delta x_2^t - (n+1)\eta \Delta x_1^{t-m} + n\eta \Delta x_1^{t-m-1}\right\} \tag{A14}$$

$$= \underbrace{(\Delta x_1^t - i\Delta x_2^t)}_{=e^{\lambda_t+i\theta_t}} + i(n+1)\eta \underbrace{(\Delta x_1^{t-m} - i\Delta x_2^{t-m})}_{=e^{\lambda_{t-m}+i\theta_{t-m}}} - in\eta \underbrace{(\Delta x_1^{t-m-1} - i\Delta x_2^{t-m-1})}_{=e^{\lambda_{t-m-1}+i\theta_{t-m-1}}} \tag{A15}$$

$$= e^{\lambda_t+i\theta_t} + i(n+1)\eta e^{\lambda_{t-m}+i\theta_{t-m}} - in\eta e^{\lambda_{t-m-1}+i\theta_{t-m-1}}. \tag{A16}$$

Let us prove Eq. (A10) by induction. Assume Eqs. (A10) up to $(\lambda_t, \theta_t)$, and then the time delay is evaluated as

$$e^{\lambda_{t-m}+i\theta_{t-m}} = e^{\lambda_t - m(m-n+\frac{1}{2})\eta^2 + \Theta(\eta^4)} e^{i(\theta_t - m\eta + \Theta(\eta^3))} \tag{A17}$$

$$= e^{\lambda_t}(1 + \Theta(\eta^2))e^{i\theta_t}(1 - im\eta + \Theta(\eta^2) + i\Theta(\eta^3)) \tag{A18}$$

$$= e^{\lambda_t+i\theta_t}(1 - im\eta + \Theta(\eta^2) + i\Theta(\eta^3)). \tag{A19}$$

Using this, we obtain the LHS of Eq. (A16) from the RHS as

$$e^{\lambda_{t+1}+i\theta_{t+1}} = e^{\lambda_t+i\theta_t} + i(n+1)\eta e^{\lambda_{t-m}-i\theta_{t-m}} + in\eta e^{\lambda_{t-m-1}+i\theta_{t-m-1}} \tag{A20}$$

$$= e^{\lambda_t+i\theta_t} + i(n+1)\eta e^{\lambda_t+i\theta_t}(1 - im\eta + \Theta(\eta^2) + i\Theta(\eta^3))$$
$$- in\eta e^{\lambda_t+i\theta_t}(1 - i(m+1)\eta + \Theta(\eta^2) + i\Theta(\eta^3)) \tag{A21}$$

$$= e^{\lambda_t+i\theta_t}(1 + i\eta + (m-n)\eta^2 + i\Theta(\eta^3) + \Theta(\eta^4)) \tag{A22}$$

$$= e^{\lambda_t+i\theta_t}e^{i\eta+(m-n+\frac{1}{2})\eta^2+i\Theta(\eta^3)+\Theta(\eta^4)} \tag{A23}$$

$$= e^{\{\lambda_t+(m-n+\frac{1}{2})\eta^2+\Theta(\eta^4)\}+i\{\theta_t+\eta+\Theta(\eta^3)\}}. \tag{A24}$$

By comparing the real and imaginary parts in Eq. (A24), we obtain

$$\lambda_{t+1} = \lambda_t + \left(m - n + \frac{1}{2}\right)\eta^2 + \Theta(\eta^4), \quad \theta_{t+1} = \theta_t + \eta + \Theta(\eta^3), \tag{A25}$$

corresponding to Eq. (A10) by rescaling the learning rate $\eta \leftarrow 2\eta$. Because we derived Eq. (A10) for time $t+1$ under the assumption of Eq. (A10) until time $t$, we have proved the lemma by induction. $\square$

## B Proof of Thm. 6

*Proof.* To introduce meaningful regret in the unconstrained setting, we assume that $\boldsymbol{x}_i \in \mathbb{R}^d$ is bounded as

$$\|\boldsymbol{x}_i\|_1 \leq \frac{1}{T}\sum_{t=1}^T \|\boldsymbol{x}_i^t\|_1. \tag{A26}$$

However, this bound is not essential in the following discussion. Now, social regret is lower-bounded as

$$\text{REGTOT}(T) = \max_{\boldsymbol{x}_i} \sum_{i=1,2}\sum_{t=1}^T \langle \boldsymbol{x}_i - \boldsymbol{x}_i^t, \boldsymbol{u}_i^t \rangle \tag{A27}$$

$$= \max_{\boldsymbol{x}_1}\sum_{t=1}^T \underbrace{\langle \boldsymbol{x}_1, \boldsymbol{u}_1^t \rangle}_{=\langle \boldsymbol{x}_1, \mathbf{c}\rangle\Delta x_2^t} + \max_{\boldsymbol{x}_2}\sum_{t=1}^T \underbrace{\langle \boldsymbol{x}_2, \boldsymbol{u}_2^t \rangle}_{=\langle \boldsymbol{x}_2, \mathbf{c}\rangle\Delta x_1^t} - \underbrace{\sum_{t=1}^T\sum_{i=1,2}\langle \boldsymbol{x}_i^t, \boldsymbol{u}_i^t \rangle}_{=0} \tag{A28}$$

$$= \max_{\boldsymbol{x}_1}\langle \boldsymbol{x}_1, \mathbf{c}\rangle\sum_{t=1}^T \Delta x_2^t + \max_{\boldsymbol{x}_2}\langle \boldsymbol{x}_2, \mathbf{c}\rangle\sum_{t=1}^T \Delta x_1^t \tag{A29}$$

$$= \|\boldsymbol{x}_1\|_1\left|\sum_{t=1}^T \Delta x_2^t\right| + \|\boldsymbol{x}_2\|_1\left|\sum_{t=1}^T \Delta x_1^t\right| \tag{A30}$$

$$\geq D\sum_{i=1,2}\left|\sum_{t=1}^T \Delta x_i^t\right| \tag{A31}$$

$$\geq D\sqrt{\sum_{i=1,2}\left|\sum_{t=1}^T \Delta x_i^t\right|^2}. \tag{A32}$$

Here, we define the scale of their strategy space $D := \min_{i=1,2}\|\boldsymbol{x}_i\|_1$.

By Lem. A1, we immediately obtain

$$\lambda_t = \lambda_1 + \underbrace{\left(m - n + \frac{1}{2}\right)\eta^2(t-1)}_{=:\alpha} + \Theta(\eta^4(t-1)), \quad \theta_t = \theta_1 + \eta(t-1) + \Theta(\eta^3(t-1)).$$

$$\tag{A33}$$

Using this, we obtain

$$\sum_{t=1}^{T} \Delta x_1^t - i \sum_{t=1}^{T} \Delta x_2^t = \sum_{t=1}^{T} e^{\lambda_t + i\theta_t} \tag{A34}$$

$$= \frac{e^{\lambda_1 + i\theta_1} - e^{\lambda_{T+1} + i\theta_{T+1}}}{1 - e^{i\eta + \alpha\eta^2 + i\Theta(\eta^3) + \Theta(\eta^4)}} \tag{A35}$$

$$= e^{\lambda_1 + i\theta_1} \frac{1 - e^{(\alpha\eta^2 + i\eta)T + i\Theta(\eta^3 T) + \Theta(\eta^4 T)}}{1 - e^{\alpha\eta^2 + i\eta + i\Theta(\eta^3) + \Theta(\eta^4)}}. \tag{A36}$$

Here, we used

$$e^{\lambda_1 + i\theta_1} - e^{\lambda_{T+1} + i\theta_{T+1}} = \sum_{t=1}^{T} (e^{\lambda_t + i\theta_t} - e^{\lambda_{t+1} + i\theta_{t+1}}) \tag{A37}$$

$$= \sum_{t=1}^{T} e^{\lambda_t + i\theta_t} \left(1 - e^{i\eta + \alpha\eta^2 + i\Theta(\eta^3) + \Theta(\eta^4)}\right) \tag{A38}$$

$$= \left(1 - e^{i\eta + \alpha\eta^2 + i\Theta(\eta^3) + \Theta(\eta^4)}\right) \sum_{t=1}^{T} e^{\lambda_t + i\theta_t}. \tag{A39}$$

Finally, we evaluate

$$\sum_{i=1,2} \left| \sum_{t=1}^{T} \Delta x_i^t \right|^2 = \left( \sum_{t=1}^{T} \Delta x_1^t - i \sum_{t=1}^{T} \Delta x_2^t \right) \left( \sum_{t=1}^{T} \Delta x_1^t + i \sum_{t=1}^{T} \Delta x_2^t \right) \tag{A40}$$

$$= \frac{e^{\lambda_1 + i\theta_1} - e^{\lambda_{T+1} + i\theta_{T+1}}}{1 - e^{i\eta + \alpha\eta^2 + i\Theta(\eta^3) + \Theta(\eta^4)}} \frac{e^{\lambda_1 - i\theta_1} - e^{\lambda_{T+1} - i\theta_{T+1}}}{1 - e^{-i\eta + \alpha\eta^2 - i\Theta(\eta^3) + \Theta(\eta^4)}} \tag{A41}$$

$$= e^{\lambda_1 + i\theta_1} \frac{1 - e^{(i\eta + \alpha\eta^2)T + i\Theta(\eta^3 T) + \Theta(\eta^4 T)}}{1 - e^{i\eta + \alpha\eta^2 + i\Theta(\eta^3) + \Theta(\eta^4)}}$$

$$\times e^{\lambda_1 - i\theta_1} \frac{1 - e^{(-i\eta + \alpha\eta^2)T - i\Theta(\eta^3 T) + \Theta(\eta^4 T)}}{1 - e^{-i\eta + \alpha\eta^2 - i\Theta(\eta^3) + \Theta(\eta^4)}} \tag{A42}$$

$$= e^{2\lambda_1} \frac{1 - 2e^{\alpha\eta^2 T + \Theta(\eta^4 T)} \cos(\eta T + \Theta(\eta^3 T)) + e^{2\alpha\eta^2 T + \Theta(\eta^4 T)}}{1 - 2e^{\alpha\eta^2 + \Theta(\eta^4)} \cos(\eta + \Theta(\eta^3)) + e^{2\alpha\eta^2 + \Theta(\eta^4)}} \tag{A43}$$

$$= e^{2\lambda_1} \frac{1 + \Theta(1)e^{\alpha\eta^2 T + \Theta(\eta^4 T)} + e^{2\alpha\eta^2 T + \Theta(\eta^4 T)}}{\eta^2 + \Theta(\eta^4)} \tag{A44}$$

$$= \Omega\left(\frac{\max\{1, e^{2\alpha\eta^2 T}\}}{\eta^2}\right) \tag{A45}$$

$$\geq \Omega(T). \tag{A46}$$

Here, we used

$$1 - 2e^{\alpha\eta^2 + \Theta(\eta^4)} \cos(\eta + \Theta(\eta^3)) + e^{2\alpha\eta^2 + \Theta(\eta^4)} \tag{A47}$$

$$= 1 - 2\left(1 + \alpha\eta^2 + \Theta(\eta^4)\right)\left(1 - \frac{1}{2}\eta^2 + \Theta(\eta^4)\right) + \left(1 + 2\alpha\eta^2 + \Theta(\eta^4)\right) \tag{A48}$$

$$= \eta^2 + \Theta(\eta^4). \tag{A49}$$

The equality holds when and only when $\eta = \Omega(1/\sqrt{T})$. In conclusion, we obtain the lower bound of social regret as $\text{REGTOT}(T) \geq \Omega(D\sqrt{T})$, leading to $\text{REGTOT}(T) \geq \Omega(\sqrt{T})$ by ignoring the scale of the strategy space. $\square$

## C   Proof of Thm. 7

*Proof.* In Matching Pennies, the Nash equilibrium $(\boldsymbol{x}_1^*, \boldsymbol{x}_2^*)$ satisfies

$$\boldsymbol{A}\boldsymbol{x}_1^* = \boldsymbol{A}^{\mathrm{T}}\boldsymbol{x}_2^* = \boldsymbol{0} \iff \Delta x_1^* = \Delta x_2^* = 0. \tag{A50}$$

We discuss the distance from the Nash equilibria, which is formulated as

$$\text{DIS}(T) = \min_{\boldsymbol{x}^*} \frac{1}{2} \sum_{i=1,2} \|\boldsymbol{x}_i^T - \boldsymbol{x}_i^*\|_2^2 \overset{\text{a}}{=} \frac{1}{2} \sum_{i=1,2} \frac{1}{4} |\Delta x_i^T|^2 \underbrace{\|\mathbf{c}\|_2^2}_{=2} = \frac{1}{4} \sum_{i=1,2} |\Delta x_i^T|^2 \qquad \text{(A51)}$$

$$\overset{\text{b}}{=} \frac{1}{4} e^{\lambda_T} = \frac{1}{4} e^{\lambda_0 + \alpha \eta^2 T + \Theta(\eta^4 T)}. \qquad \text{(A52)}$$

In (a), we used the nearest Nash equilibrium from $(\boldsymbol{x}_1, \boldsymbol{x}_2)$ as

$$\arg\min_{\boldsymbol{x}_i^*} \|\boldsymbol{x}_i - \boldsymbol{x}_i^*\|_2^2 = \boldsymbol{x}_i - \frac{1}{2} \Delta x_i \mathbf{c}. \qquad \text{(A53)}$$

In (b), we applied Lem. A1. For OFTRL $n = 1$ with a time delay $m \geq 1$, $\alpha = n - m + \frac{1}{2} > 0$ holds. For sufficiently small $\eta$, the term of $\Theta(\eta^4 T)$ is negligible, and $D^T$ diverges with the final time $T$. We have proved Thm. 7. $\qquad \square$

## D  Proof of Thm. 10

*Proof.* First, for all $\tilde{\mathbf{f}} \in \mathbb{R}^d$, we define $\mathbf{f}$ and $\mathrm{F}$ as

$$\mathbf{f} := \arg\max_{\boldsymbol{x}_i \in \mathcal{X}_i} \eta \langle \boldsymbol{x}_i, \tilde{\mathbf{f}} \rangle - h(\boldsymbol{x}_i), \quad \mathrm{F} := \max_{\boldsymbol{x}_i \in \mathcal{X}_i} \eta \langle \boldsymbol{x}_i, \tilde{\mathbf{f}} \rangle - h(\boldsymbol{x}_i). \qquad \text{(A54)}$$

In this notation, we define

$$\tilde{\boldsymbol{g}}_i^t := \sum_{s=1}^{t} \boldsymbol{u}_i^s, \quad \tilde{\boldsymbol{x}}_i^t := \sum_{s=1}^{t-m-1} \boldsymbol{u}_i^s + (m+1)\boldsymbol{u}_i^{t-m-1}. \qquad \text{(A55)}$$

Here, $\tilde{\boldsymbol{g}}_i^t$ represents $i$'s cumulative rewards until time $t$. Also, $\tilde{\boldsymbol{x}}_i^t$ is a prediction of $\tilde{\boldsymbol{g}}_i^t$ by delayed feedback because all the unobservable rewards $\boldsymbol{u}_i^{t-m}, \cdots, \boldsymbol{u}_i^t$ are replaced by the latest observable reward $\boldsymbol{u}_i^{t-m-1}$. According to $\tilde{\boldsymbol{g}}_i^t$ and $\tilde{\boldsymbol{x}}_i^t$, we use $(\boldsymbol{g}_i^t, G_i^t)$ and $(\boldsymbol{x}_i^t, X_i^t)$. Note that $\boldsymbol{x}_i^t$ corresponds to the WOFTRL algorithm for the case of $n = m + 1$.

Now, between different $\tilde{\mathbf{f}}, \tilde{\mathbf{f}}' \in \mathbb{R}^d$, the following lemma holds.

**Lemma A2** (First-order optimality condition). *For all $\tilde{\mathbf{f}}, \tilde{\mathbf{f}}' \in \mathbb{R}^d$ and their corresponding $(\mathbf{f}, \mathrm{F}), (\mathbf{f}', \mathrm{F}')$, the following inequalities hold*

$$\frac{1}{2} \|\mathbf{f}' - \mathbf{f}\|_2^2 \leq -\mathrm{F}' + \mathrm{F} + \eta \langle \mathbf{f}', \tilde{\mathbf{f}}' - \tilde{\mathbf{f}} \rangle, \qquad \text{(FO1)}$$

$$\|\mathbf{f}' - \mathbf{f}\|_2^2 \leq \eta \langle \mathbf{f}' - \mathbf{f}, \tilde{\mathbf{f}}' - \tilde{\mathbf{f}} \rangle, \qquad \text{(FO2)}$$

$$\|\mathbf{f}' - \mathbf{f}\|_2 \leq \eta \|\tilde{\mathbf{f}}' - \tilde{\mathbf{f}}\|_2. \qquad \text{(FO3)}$$

Now, the payoff at time $t$ can be divided as follows;

$$-\langle \boldsymbol{x}_i^t, \boldsymbol{u}_i^t \rangle = \langle \boldsymbol{g}_i^t - \boldsymbol{x}_i^t, \tilde{\boldsymbol{g}}_i^t - \tilde{\boldsymbol{x}}_i^t \rangle - \langle \boldsymbol{x}_i^t, \tilde{\boldsymbol{x}}_i^t - \tilde{\boldsymbol{g}}_i^{t-1} \rangle - \langle \boldsymbol{g}_i^t, \tilde{\boldsymbol{g}}_i^t - \tilde{\boldsymbol{x}}_i^t \rangle. \qquad \text{(A56)}$$

The first term is upper-bounded as

$$\sum_{t=1}^{T} \left\langle \boldsymbol{g}_i^t - \boldsymbol{x}_i^t, \tilde{\boldsymbol{g}}_i^t - \tilde{\boldsymbol{x}}_i^t \right\rangle \tag{A57}$$

$$\leq \sum_{t=1}^{T} \|\boldsymbol{g}_i^t - \boldsymbol{x}_i^t\|_2 \|\tilde{\boldsymbol{g}}_i^t - \tilde{\boldsymbol{x}}_i^t\|_2 \tag{A58}$$

$$\leq \eta \sum_{t=1}^{T} \|\tilde{\boldsymbol{g}}_i^t - \tilde{\boldsymbol{x}}_i^t\|_2^2 \tag{A59}$$

$$= \eta \sum_{t=1}^{T} \left\| \sum_{s=t-m}^{t} \boldsymbol{u}_i^s - (m+1)\boldsymbol{u}_i^{t-m-1} \right\|_2^2 \tag{A60}$$

$$= \eta \sum_{t=1}^{T} \left\| \sum_{s=t-m}^{t} (t-s+1)(\boldsymbol{u}_i^s - \boldsymbol{u}_i^{s-1}) \right\|_2^2 \tag{A61}$$

$$\leq \eta \sum_{t=1}^{T} \left( \sum_{s=t-m}^{t} (t-s+1)\|\boldsymbol{u}_i^s - \boldsymbol{u}_i^{s-1}\|_2 \right)^2 \tag{A62}$$

$$= \eta \sum_{t=1}^{T} \sum_{s=t-m}^{t} (t-s+1) \sum_{s'=t-m}^{t} (t-s'+1)\|\boldsymbol{u}_i^s - \boldsymbol{u}_i^{s-1}\|_2 \|\boldsymbol{u}_i^{s'} - \boldsymbol{u}_i^{s'-1}\|_2 \tag{A63}$$

$$\leq \eta \sum_{t=1}^{T} \sum_{s=t-m}^{t} (t-s+1) \sum_{s'=t-m}^{t} (t-s'+1)\frac{1}{2} \left( \|\boldsymbol{u}_i^s - \boldsymbol{u}_i^{s-1}\|_2^2 + \|\boldsymbol{u}_i^{s'} - \boldsymbol{u}_i^{s'-1}\|_2^2 \right) \tag{A64}$$

$$\leq \eta \sum_{t=1}^{T} \sum_{s=t-m}^{t} (t-s+1) \sum_{s'=t-m}^{t} (t-s'+1)\|\boldsymbol{u}_i^s - \boldsymbol{u}_i^{s-1}\|_2^2 \tag{A65}$$

$$= \lambda\eta \sum_{t=1}^{T} \sum_{s=t-m}^{t} (t-s+1)\|\boldsymbol{u}_i^s - \boldsymbol{u}_i^{s-1}\|_2^2 \tag{A66}$$

$$\leq \lambda^2\eta \sum_{t=1}^{T} \|\boldsymbol{u}_i^t - \boldsymbol{u}_i^{t-1}\|_2^2. \tag{A67}$$

In the final two lines, we twice used

$$\sum_{s=t-m}^{t} (t-s+1) = \frac{(m+1)(m+2)}{2} =: \lambda. \tag{A68}$$

Furthermore, the second and third terms are also upper-bounded as

$$-\sum_{t=1}^{T} \left\langle \boldsymbol{x}_i^t, \tilde{\boldsymbol{x}}_i^t - \tilde{\boldsymbol{g}}_i^{t-1} \right\rangle - \sum_{t=1}^{T} \left\langle \boldsymbol{g}_i^t, \tilde{\boldsymbol{g}}_i^t - \tilde{\boldsymbol{x}}_i^t \right\rangle \tag{A69}$$

$$\overset{\mathrm{a}}{\leq} \sum_{t=1}^{T} \frac{1}{\eta} \left( -X_i^t + G_i^{t-1} - \frac{1}{2}\|\boldsymbol{x}_i^t - \boldsymbol{g}_i^{t-1}\|_2^2 \right) + \sum_{t=1}^{T} \frac{1}{\eta} \left( -G_i^t + X_i^t - \frac{1}{2}\|\boldsymbol{g}_i^t - \boldsymbol{x}_i^t\|_2^2 \right) \tag{A70}$$

$$= \frac{-G_i^T + G_i^0}{\eta} - \frac{1}{2\eta} \sum_{t=1}^{T} (\|\boldsymbol{x}_i^t - \boldsymbol{g}_i^{t-1}\|_2^2 + \|\boldsymbol{g}_i^t - \boldsymbol{x}_i^t\|_2^2) \tag{A71}$$

$$\overset{\mathrm{b}}{\leq} -\max_{\boldsymbol{x}_i \in \mathcal{X}_i} \left\langle \boldsymbol{x}_i, \sum_{t=1}^{T} \boldsymbol{u}_i^t \right\rangle + \frac{h_{\max}}{\eta} - \frac{1}{2\eta} \sum_{t=1}^{T} (\|\boldsymbol{x}_i^t - \boldsymbol{g}_i^{t-1}\|_2^2 + \|\boldsymbol{g}_i^t - \boldsymbol{x}_i^t\|_2^2) \tag{A72}$$

$$\overset{\mathrm{c}}{\leq} -\max_{\boldsymbol{x}_i \in \mathcal{X}_i} \left\langle \boldsymbol{x}_i, \sum_{t=1}^{T} \boldsymbol{u}_i^t \right\rangle + \frac{h_{\max}}{\eta} - \frac{1}{4\eta} \sum_{t=1}^{T} \|\boldsymbol{x}_i^t - \boldsymbol{x}_i^{t-1}\|_2^2. \tag{A73}$$

In (a), we used Eq. (FO1) for $(\tilde{\mathbf{f}}, \tilde{\mathbf{f}}') = (\tilde{\boldsymbol{x}}_i^t, \tilde{\boldsymbol{g}}_i^{t-1})$ and $(\tilde{\boldsymbol{g}}_i^t, \tilde{\boldsymbol{x}}_i^t)$. In (b), we used

$$0 \leq G_i^0, \quad -\frac{G_i^T}{\eta} \leq -\max_{\boldsymbol{x}_i \in \mathcal{X}_i} \left\langle \boldsymbol{x}_i, \sum_{t=1}^T \boldsymbol{u}_i^t \right\rangle + \frac{h_{\max}}{\eta}, \quad h_{\max} := \max_{\boldsymbol{x}_i \in \mathcal{X}_i} h(\boldsymbol{x}_i). \tag{A74}$$

In (c), we used

$$\frac{1}{2} \sum_{t=1}^T \|\boldsymbol{x}_i^t - \boldsymbol{x}_i^{t-1}\|^2 \leq \sum_{t=1}^T (\|\boldsymbol{x}_i^t - \boldsymbol{g}_i^{t-1}\|^2 + \|\boldsymbol{g}_i^{t-1} - \boldsymbol{x}_i^{t-1}\|^2) \tag{A75}$$

$$\leq \sum_{t=1}^T (\|\boldsymbol{x}_i^t - \boldsymbol{g}_i^{t-1}\|^2 + \|\boldsymbol{g}_i^t - \boldsymbol{x}_i^t\|^2). \tag{A76}$$

Finally, we obtain the upper bound of the individual regret $\text{REG}_i(T)$ as

$$\text{REG}_i(T) = \max_{\boldsymbol{x}_i \in \mathcal{X}_i} \sum_{t=1}^T \left\langle \boldsymbol{x}_i - \boldsymbol{x}_i^t, \boldsymbol{u}_i^t \right\rangle \tag{A77}$$

$$= \max_{\boldsymbol{x}_i \in \mathcal{X}_i} \left\langle \boldsymbol{x}_i, \sum_{t=1}^T \boldsymbol{u}_i^t \right\rangle - \sum_{t=1}^T \left\langle \boldsymbol{x}_i^t, \boldsymbol{u}_i^t \right\rangle \tag{A78}$$

$$\leq \frac{h_{\max}}{\eta} + \lambda^2 \eta \sum_{t=1}^T \|\boldsymbol{u}_i^t - \boldsymbol{u}_i^{t-1}\|_2^2 - \frac{1}{4\eta} \sum_{t=1}^T \|\boldsymbol{x}_i^t - \boldsymbol{x}_i^{t-1}\|_2^2. \tag{A79}$$

Thus, we have proven that the regret satisfies the RVU property with $\alpha = h_{\max}/\eta$, $\beta = \lambda^2 \eta$, and $\gamma = 1/(4\eta)$. $\square$

### D.1 Proof of Lem. A2

*Proof.*

$$\text{F} = \eta \langle \mathbf{f}, \tilde{\mathbf{f}} \rangle - h(\mathbf{f}) \tag{A80}$$

$$\geq \eta \langle \mathbf{f}', \tilde{\mathbf{f}} \rangle - h(\mathbf{f}') + \frac{1}{2}\|\mathbf{f}' - \mathbf{f}\|_2^2 \tag{A81}$$

$$= \eta \langle \mathbf{f}', \tilde{\mathbf{f}}' \rangle - h(\mathbf{f}') - \eta \langle \mathbf{f}', \tilde{\mathbf{f}}' - \tilde{\mathbf{f}} \rangle + \frac{1}{2}\|\mathbf{f}' - \mathbf{f}\|_2^2 \tag{A82}$$

$$= \text{F}' - \eta \langle \mathbf{f}', \tilde{\mathbf{f}}' - \tilde{\mathbf{f}} \rangle + \frac{1}{2}\|\mathbf{f}' - \mathbf{f}\|_2^2. \tag{A83}$$

In the inequality, we used 1-strict convexity of $h$. This corresponds to Eq. (FO1).

Furthermore, by summing up

$$\left( \frac{1}{2}\|\mathbf{f}' - \mathbf{f}\|_2^2 \leq -\text{F}' + \text{F} + \eta \langle \mathbf{f}', \tilde{\mathbf{f}}' - \tilde{\mathbf{f}} \rangle, \quad \frac{1}{2}\|\mathbf{f} - \mathbf{f}'\|_2^2 \leq -\text{F} + \text{F}' + \eta \langle \mathbf{f}, \tilde{\mathbf{f}} - \tilde{\mathbf{f}}' \rangle \right) \tag{A84}$$

$$\Rightarrow \|\mathbf{f}' - \mathbf{f}\|_2^2 \leq \eta \langle \mathbf{f}' - \mathbf{f}, \tilde{\mathbf{f}}' - \tilde{\mathbf{f}} \rangle. \tag{A85}$$

This corresponds to Eq. (FO2).

Finally, by the Cauchy–Schwarz inequality, we obtain

$$\|\mathbf{f}' - \mathbf{f}\|_2^2 \leq \eta \langle \mathbf{f}' - \mathbf{f}, \tilde{\mathbf{f}}' - \tilde{\mathbf{f}} \rangle \leq \eta \|\mathbf{f}' - \mathbf{f}\|_2 \|\tilde{\mathbf{f}}' - \tilde{\mathbf{f}}\|_2 \tag{A86}$$

$$\Rightarrow \|\mathbf{f}' - \mathbf{f}\|_2 \leq \eta \|\tilde{\mathbf{f}}' - \tilde{\mathbf{f}}\|_2. \tag{A87}$$

This corresponds to Eq. (FO3). $\square$

## E  Proof of Cor. 13

*Proof.* We prove the last-iterate convergence by the following five steps.

**Step 1: Equivalence between generalized FTRL and generalized MD**   When $h$ is a Legendre function, the strategies always exist in the interior of the strategy space, i.e., $\boldsymbol{x}^t \in \text{int}(\mathcal{X})$ for all $t$. Furthermore, the dynamics of generalized FTRL become equivalent to those of generalized MD as follows.

**Definition A3** (Generalized Mirror Descent). *With the time delay of $m \in \mathbb{N}$ steps, generalized Mirror Descent is formulated based on the prox-mapping $\boldsymbol{P}$ as follows*

$$\boldsymbol{x}^{t+1} = \boldsymbol{P}(\hat{\boldsymbol{x}}^{t-m+1}, \boldsymbol{m}^t), \quad \hat{\boldsymbol{x}}^{t+1} = \boldsymbol{P}(\hat{\boldsymbol{x}}^t, \boldsymbol{u}^t), \quad \boldsymbol{P}(\hat{\boldsymbol{x}}, \boldsymbol{u}) := \arg\max_{\boldsymbol{x} \in \mathcal{X}} \eta \langle \boldsymbol{x}, \boldsymbol{u} \rangle - D(\boldsymbol{x}, \hat{\boldsymbol{x}}),$$
$$\text{(A88)}$$

*where $D(\boldsymbol{x}, \boldsymbol{x}')$ is the Bregman divergence between $\boldsymbol{x}, \boldsymbol{x}' \in \mathcal{X}$, defined as*

$$D(\boldsymbol{x}, \boldsymbol{x}') := h(\boldsymbol{x}) - \langle \boldsymbol{x} - \boldsymbol{x}', \nabla h(\boldsymbol{x}') \rangle - h(\boldsymbol{x}'). \quad \text{(A89)}$$

**Lemma A4** (Equivalence between generalized FTRL and generalized MD). *Suppose that $h$ is a Legendre function, then the time series of $\{\boldsymbol{x}^t\}_{t=1,\dots}$ are equivalent between Defs. 2 and A3.*

**Step 2: Existence of Lyapunov function**   In accordance with WOFTRL, suppose $\boldsymbol{m}^t = n\boldsymbol{u}^{t-m}$ for $n = m + 1$ in generalized MD, then there exists a Lyapunov function, which is defined for all $\boldsymbol{x}^* \in \mathcal{X}^*$ as

$$V^t(\boldsymbol{x}^*) := D(\boldsymbol{x}^*, \hat{\boldsymbol{x}}^t) + \frac{1}{2}\Big\{D(\hat{\boldsymbol{x}}^t, \boldsymbol{x}^{t-1}) + \sum_{r=1}^{n-1} \frac{n-r}{n}(D(\boldsymbol{x}^{t-r}, \hat{\boldsymbol{x}}^{t-r}) + D(\hat{\boldsymbol{x}}^{t-r}, \boldsymbol{x}^{t-r-1}))\Big\}.$$
$$\text{(A90)}$$

For $\eta \leq 1/(\sqrt{8}n^2 L)$, we see that this $V^t(\boldsymbol{x}^*)$ monotonically decreases for all $\boldsymbol{x}^* \in \mathcal{X}^*$, as follows.

**Lemma A5** (Lyapunov function under time delay). *Suppose generalized MD with $n = m + 1$ and $\eta \leq 1/(\sqrt{8}n^2 L)$, then for all $\boldsymbol{x}^* \in \mathcal{X}^*$, $V^t(\boldsymbol{x}^*)$ is monotonic decreasing as*

$$V^{t+1}(\boldsymbol{x}^*) \leq V^t(\boldsymbol{x}^*) - \frac{1}{2}D(\hat{\boldsymbol{x}}^{t+1}, \boldsymbol{x}^t) - \frac{1}{2}D(\boldsymbol{x}^t, \hat{\boldsymbol{x}}^t). \quad \text{(A91)}$$

By summing up Eq. (A91), for all $\boldsymbol{x}^* \in \mathcal{X}^*$, we derive

$$V^1(\boldsymbol{x}^*) \geq \lim_{T \to \infty}\left[V^T(\boldsymbol{x}^*) + \frac{1}{2}\sum_{t=1}^T (D(\hat{\boldsymbol{x}}^{t+1}, \boldsymbol{x}^t) + D(\boldsymbol{x}^t, \hat{\boldsymbol{x}}^t))\right] \quad \text{(A92)}$$

$$\geq \frac{1}{2}\sum_{t=1}^{\infty} (D(\hat{\boldsymbol{x}}^{t+1}, \boldsymbol{x}^t) + D(\boldsymbol{x}^t, \hat{\boldsymbol{x}}^t)) \quad \text{(A93)}$$

$$\geq \frac{1}{4}\sum_{t=1}^{\infty} (\|\hat{\boldsymbol{x}}^{t+1} - \boldsymbol{x}^t\|_2^2 + \|\boldsymbol{x}^t - \hat{\boldsymbol{x}}^t\|_2^2) \quad \text{(A94)}$$

$$\geq \frac{1}{8}\sum_{t=1}^{\infty} (\|\hat{\boldsymbol{x}}^{t+1} - \hat{\boldsymbol{x}}^t\|_2^2). \quad \text{(A95)}$$

By Eqs. (A94) and (A95), we obtain

$$\lim_{t \to \infty} \|\hat{\boldsymbol{x}}^{t+1} - \boldsymbol{x}^t\|_2 = \lim_{t \to \infty} \|\hat{\boldsymbol{x}}^{t+1} - \hat{\boldsymbol{x}}^t\|_2 = 0. \quad \text{(A96)}$$

**Step 3: Existence of subsequence converging to equilibrium**   Since $\mathcal{X}$ is a compact space, the Bolzano-Weierstrass theorem can be applied, and there exists a subsequence $\{\boldsymbol{x}^{t_l}\}_{l=1,\dots}$ equipped with its limit. In other words, there exists $\tilde{\boldsymbol{x}} \in \mathcal{X}$ such that

$$\lim_{l \to \infty} \boldsymbol{x}^{t_l} = \tilde{\boldsymbol{x}}. \quad \text{(A97)}$$

By using Eqs. (A96), we further obtain

$$\lim_{l \to \infty} \hat{\boldsymbol{x}}^{t_l+1} = \tilde{\boldsymbol{x}}, \quad \lim_{l \to \infty} \hat{\boldsymbol{x}}^{t_l} = \tilde{\boldsymbol{x}}. \quad \text{(A98)}$$

Thus, by substituting Eqs. (A97) and (A98) into the prox-mapping (the second one in Eqs. (A88)), we show that this $\tilde{\boldsymbol{x}}$ satisfies the fixed point condition of the prox-mapping, i.e.,

$$\tilde{\boldsymbol{x}} = \lim_{l \to \infty} \hat{\boldsymbol{x}}^{t_l+1} = \lim_{l \to \infty} \boldsymbol{P}(\hat{\boldsymbol{x}}^{t_l}, \boldsymbol{u}(\boldsymbol{x}^{t_l})) \stackrel{\text{a}}{=} \boldsymbol{P}\left(\lim_{l \to \infty} \hat{\boldsymbol{x}}^{t_l}, \lim_{l \to \infty} \boldsymbol{u}(\boldsymbol{x}^{t_l})\right) = \boldsymbol{P}(\tilde{\boldsymbol{x}}, \boldsymbol{u}(\tilde{\boldsymbol{x}})). \quad \text{(A99)}$$

In (a), we used the continuity of prox-mapping $\boldsymbol{P}$, as the following lemma shows.

**Lemma A6** (Continuity of prox-mapping). *The prox-mapping $\boldsymbol{P}(\hat{\boldsymbol{x}}, \boldsymbol{u})$ is continuous for $\hat{\boldsymbol{x}} \in \text{int}(\mathcal{X})$ and $\boldsymbol{u} \in \mathbb{R}^d$.*

By the first-order optimality condition, this fixed point satisfies the Nash equilibrium condition [10], i.e., $\tilde{\boldsymbol{x}} \in \mathcal{X}^*$. In conclusion, we obtain a subsequence $\{\boldsymbol{x}^{t_l}\}_{l=1,\dots}$ which converges to one Nash equilibrium $\boldsymbol{x}^* \in \mathcal{X}^*$.

**Step 4: Convergence guarantee by Lyapunov function** Let $\boldsymbol{x}^*$ denote the specific Nash equilibrium, to which the subsequence $\{t_l\}_{l=1,\dots}$ converges. Because $V^t(\boldsymbol{x}^*)$ is both lower-bounded by $0$ and monotonic decreasing, there exists its limits which corresponds to its limit inferior, and we obtain

$$0 \leq \lim_{t\to\infty} V^t(\boldsymbol{x}^*) = \liminf_{t\to\infty} V^t(\boldsymbol{x}^*) \leq \lim_{l\to\infty} V^{t_l}(\boldsymbol{x}^*) = 0. \tag{A100}$$

This trivially leads to

$$\lim_{t\to\infty} D(\boldsymbol{x}^*, \hat{\boldsymbol{x}}^t) \leq \lim_{t\to\infty} V^t(\boldsymbol{x}^*) = 0 \Rightarrow \lim_{t\to\infty} \hat{\boldsymbol{x}}^t = \boldsymbol{x}^*. \tag{A101}$$

By further using $\lim_{t\to\infty} \|\hat{\boldsymbol{x}}^{t+1} - \boldsymbol{x}^t\|_2 = 0$, we ultimately obtain

$$\lim_{t\to\infty} \boldsymbol{x}^t = \boldsymbol{x}^*. \tag{A102}$$

indicating the last-iterate convergence to one of the Nash equilibria. □

### E.1 Proof of Lem. A4

*Proof.* First, GFTRL is obviously rewritten as

$$\boldsymbol{x}^{t+1} = \arg\max_{\boldsymbol{x}\in\mathcal{X}} \eta\langle \boldsymbol{x}, \boldsymbol{m}^t\rangle - F(\boldsymbol{x}, \tilde{\boldsymbol{g}}^{t-m+1}). \tag{A103}$$

Here, $F(\boldsymbol{x}, \tilde{\boldsymbol{g}})$ is the Fenchel coupling between $\boldsymbol{x} \in \mathcal{X}$ and $\tilde{\boldsymbol{g}} \in \mathbb{R}^d$, which is defined as

$$F(\boldsymbol{x}, \tilde{\boldsymbol{g}}) := h(\boldsymbol{x}) - \langle \boldsymbol{x}, \tilde{\boldsymbol{g}}\rangle + h^*(\tilde{\boldsymbol{g}}), \quad h^*(\tilde{\boldsymbol{g}}) := \max_{\boldsymbol{x}\in\mathcal{X}} \langle \boldsymbol{x}, \tilde{\boldsymbol{g}}\rangle - h(\boldsymbol{x}). \tag{A104}$$

Hence, it is sufficient to see the equivalence between $D(\boldsymbol{x}, \hat{\boldsymbol{x}}^t)$ and $F(\boldsymbol{x}, \tilde{\boldsymbol{g}}^t)$, which is proven as

$$D(\boldsymbol{x}, \hat{\boldsymbol{x}}^t) = h(\boldsymbol{x}) - \langle \boldsymbol{x} - \hat{\boldsymbol{x}}^t, \nabla h(\hat{\boldsymbol{x}}^t)\rangle + h(\hat{\boldsymbol{x}}^t) \tag{A105}$$

$$\stackrel{\text{a}}{=} h(\boldsymbol{x}) - \langle \boldsymbol{x} - \hat{\boldsymbol{x}}^t, \boldsymbol{y}^t\rangle + h(\hat{\boldsymbol{x}}^t) \tag{A106}$$

$$= h(\boldsymbol{x}) - \langle \boldsymbol{x}, \tilde{\boldsymbol{g}}^t\rangle - (\langle \hat{\boldsymbol{x}}^t, \tilde{\boldsymbol{g}}^t\rangle - h(\hat{\boldsymbol{x}}^t)) \tag{A107}$$

$$\stackrel{\text{b}}{=} h(\boldsymbol{x}) - \langle \boldsymbol{x}, \tilde{\boldsymbol{g}}^t\rangle - h^*(\tilde{\boldsymbol{g}}^t) \tag{A108}$$

$$= F(\boldsymbol{x}, \tilde{\boldsymbol{g}}^t). \tag{A109}$$

In (a), we applied the first-order optimality condition for $\hat{\boldsymbol{x}}^{t+1}$ in Eq. (A88) as

$$\langle \eta\boldsymbol{u}^t - \nabla h(\hat{\boldsymbol{x}}^{t+1}) + \nabla h(\hat{\boldsymbol{x}}^t), \boldsymbol{x} - \boldsymbol{x}'\rangle = 0 \tag{A110}$$

$$\Leftrightarrow \langle \nabla h(\hat{\boldsymbol{x}}^{t+1}), \boldsymbol{x} - \boldsymbol{x}'\rangle = \langle \nabla h(\hat{\boldsymbol{x}}^t) + \eta\boldsymbol{u}^t, \boldsymbol{x} - \boldsymbol{x}'\rangle \tag{A111}$$

$$\Rightarrow \langle \nabla h(\hat{\boldsymbol{x}}^{t+1}), \boldsymbol{x} - \boldsymbol{x}'\rangle = \langle \tilde{\boldsymbol{g}}^{t+1}, \boldsymbol{x} - \boldsymbol{x}'\rangle \tag{A112}$$

$$\Leftrightarrow \langle \nabla h(\hat{\boldsymbol{x}}^t), \boldsymbol{x} - \boldsymbol{x}'\rangle = \langle \tilde{\boldsymbol{g}}^t, \boldsymbol{x} - \boldsymbol{x}'\rangle, \tag{A113}$$

for all $\boldsymbol{x}, \boldsymbol{x}' \in \mathcal{X}$. This means that the projection of $\nabla h(\hat{\boldsymbol{x}}^{t+1})$ on $\mathcal{X}$ is equal to that of $\tilde{\boldsymbol{g}}^{t+1}$. In (b), we considered that $\hat{\boldsymbol{x}}^{t+1}$ satisfies the first-order optimality condition for $h^*(\tilde{\boldsymbol{g}}^{t+1})$ as

$$h^*(\boldsymbol{y}^t) = \max_{\boldsymbol{x}\in\mathcal{X}} \langle \boldsymbol{x}, \boldsymbol{y}^t\rangle - h(\boldsymbol{x}) \tag{A114}$$

$$= \langle \hat{\boldsymbol{x}}^t, \boldsymbol{y}^t\rangle - h(\hat{\boldsymbol{x}}^t). \tag{A115}$$

Thus, it has been proven that the dynamics of $\boldsymbol{x}^t$ are equivalent between generalized FTRL and generalized MD. □

## E.2 Proof of Lem. A5

*Proof.* We prove that $V^t(\boldsymbol{x}^*)$ monotonically decreases as

$$V^{t+1}(\boldsymbol{x}^*) = D(\boldsymbol{x}^*, \hat{\boldsymbol{x}}^{t+1}) + \frac{1}{2}\left\{ D(\hat{\boldsymbol{x}}^{t+1}, \boldsymbol{x}^t) + \sum_{r=1}^{n-1} \frac{n-r}{n}(D(\boldsymbol{x}^{t-r+1}, \hat{\boldsymbol{x}}^{t-r+1}) + D(\hat{\boldsymbol{x}}^{t-r+1}, \boldsymbol{x}^{t-r})) \right\} \tag{A116}$$

$$\overset{\mathrm{a}}{=} D(\boldsymbol{x}^*, \hat{\boldsymbol{x}}^t) + \frac{1}{2}\sum_{r=1}^{n-1}\frac{n-r}{n}(D(\boldsymbol{x}^{t-r+1}, \hat{\boldsymbol{x}}^{t-r+1}) + D(\hat{\boldsymbol{x}}^{t-r+1}, \boldsymbol{x}^{t-r}))$$

$$+ \left\langle \nabla h(\hat{\boldsymbol{x}}^{t+1}) - \nabla h(\boldsymbol{x}^t), \hat{\boldsymbol{x}}^{t+1} - \boldsymbol{x}^t \right\rangle - \frac{1}{2}D(\hat{\boldsymbol{x}}^{t+1}, \boldsymbol{x}^t) - D(\boldsymbol{x}^t, \hat{\boldsymbol{x}}^t) \tag{A117}$$

$$\overset{\mathrm{b}}{\leq} D(\boldsymbol{x}^*, \hat{\boldsymbol{x}}^t) + \frac{1}{2}\sum_{r=1}^{n-1}\frac{n-r}{n}(D(\boldsymbol{x}^{t-r+1}, \hat{\boldsymbol{x}}^{t-r+1}) + D(\hat{\boldsymbol{x}}^{t-r+1}, \boldsymbol{x}^{t-r}))$$

$$+ \frac{1}{2n}\sum_{r=1}^{n}(D(\boldsymbol{x}^{t-r+1}, \hat{\boldsymbol{x}}^{t-r+1}) + D(\hat{\boldsymbol{x}}^{t-r+1}, \boldsymbol{x}^{t-r})) - \frac{1}{2}D(\hat{\boldsymbol{x}}^{t+1}, \boldsymbol{x}^t) - D(\boldsymbol{x}^t, \hat{\boldsymbol{x}}^t) \tag{A118}$$

$$= D(\boldsymbol{x}^*, \hat{\boldsymbol{x}}^t) + \frac{1}{2}\left\{ D(\hat{\boldsymbol{x}}^t, \boldsymbol{x}^{t-1}) + \sum_{r=1}^{n-1}\frac{n-r}{n}(D(\boldsymbol{x}^{t-r}, \hat{\boldsymbol{x}}^{t-r}) + D(\hat{\boldsymbol{x}}^{t-r}, \boldsymbol{x}^{t-r-1})) \right\}$$

$$- \frac{1}{2}D(\hat{\boldsymbol{x}}^{t+1}, \boldsymbol{x}^t) - \frac{1}{2}D(\boldsymbol{x}^t, \hat{\boldsymbol{x}}^t) \tag{A119}$$

$$= V^t(\boldsymbol{x}^*) - \frac{1}{2}D(\hat{\boldsymbol{x}}^{t+1}, \boldsymbol{x}^t) - \frac{1}{2}D(\boldsymbol{x}^t, \hat{\boldsymbol{x}}^t). \tag{A120}$$

In (a), we used

$$D(\boldsymbol{x}^*, \hat{\boldsymbol{x}}^{t+1}) + D(\hat{\boldsymbol{x}}^{t+1}, \boldsymbol{x}^t) = D(\boldsymbol{x}^*, \hat{\boldsymbol{x}}^t) - D(\boldsymbol{x}^t, \hat{\boldsymbol{x}}^t) + \left\langle \nabla h(\hat{\boldsymbol{x}}^{t+1}) - \nabla h(\boldsymbol{x}^t), \hat{\boldsymbol{x}}^{t+1} - \boldsymbol{x}^t \right\rangle, \tag{A121}$$

which is obtained by summing up the following generalized law of cosines

$$D(\boldsymbol{x}^*, \hat{\boldsymbol{x}}^{t+1}) = D(\boldsymbol{x}^*, \hat{\boldsymbol{x}}^t) - D(\hat{\boldsymbol{x}}^{t+1}, \hat{\boldsymbol{x}}^t) + \left\langle \nabla h(\hat{\boldsymbol{x}}^{t+1}) - \nabla h(\hat{\boldsymbol{x}}^t), \hat{\boldsymbol{x}}^{t+1} - \boldsymbol{x}^* \right\rangle, \tag{A122}$$

$$D(\hat{\boldsymbol{x}}^{t+1}, \boldsymbol{x}^t) = D(\hat{\boldsymbol{x}}^{t+1}, \hat{\boldsymbol{x}}^t) - D(\boldsymbol{x}^t, \hat{\boldsymbol{x}}^t) + \left\langle \nabla h(\boldsymbol{x}^t) - \nabla h(\hat{\boldsymbol{x}}^t), \boldsymbol{x}^t - \hat{\boldsymbol{x}}^{t+1} \right\rangle. \tag{A123}$$

and using the Nash equilibrium condition

$$\left\langle \nabla h(\hat{\boldsymbol{x}}^{t+1}) - \nabla h(\hat{\boldsymbol{x}}^t), \boldsymbol{x}^* - \boldsymbol{x}^t \right\rangle = \eta \left\langle \boldsymbol{u}^t, \boldsymbol{x}^* - \boldsymbol{x}^t \right\rangle = 0. \tag{A124}$$

In (b), for $\eta \leq 1/(8\sqrt{2}n^2L)$, we used

$$\langle \nabla h(\hat{\boldsymbol{x}}^{t+1}) - \nabla h(\boldsymbol{x}^t), \hat{\boldsymbol{x}}^{t+1} - \boldsymbol{x}^t \rangle \tag{A125}$$

$$\leq \|\nabla h(\hat{\boldsymbol{x}}^{t+1}) - \nabla h(\boldsymbol{x}^t)\|^2 \tag{A126}$$

$$= \eta^2 \Big\| \sum_{r=1}^{n} \boldsymbol{u}^{t-r+1} - n\boldsymbol{u}^{t-n} \Big\|_2^2 \tag{A127}$$

$$\leq \eta^2 L^2 \Big\| \sum_{r=1}^{n} \boldsymbol{x}^{t-r+1} - n\boldsymbol{x}^{t-n} \Big\|_2^2 \tag{A128}$$

$$= \eta^2 L^2 \Big\| \sum_{r=1}^{n} \{r(\boldsymbol{x}^{t-r+1} - \hat{\boldsymbol{x}}^{t-r+1}) + r(\hat{\boldsymbol{x}}^{t-r+1} - \boldsymbol{x}^{t-r})\} \Big\|_2^2 \tag{A129}$$

$$\leq 2n\eta^2 L^2 \sum_{r=1}^{n} r^2 (\|\boldsymbol{x}^{t-r+1} - \hat{\boldsymbol{x}}^{t-r+1}\|_2^2 + \|\hat{\boldsymbol{x}}^{t-r+1} - \boldsymbol{x}^{t-r}\|_2^2) \tag{A130}$$

$$\leq 2n^3\eta^2 L^2 \sum_{r=1}^{n} (\|\boldsymbol{x}^{t-r+1} - \hat{\boldsymbol{x}}^{t-r+1}\|_2^2 + \|\hat{\boldsymbol{x}}^{t-r+1} - \boldsymbol{x}^{t-r}\|_2^2) \tag{A131}$$

$$\leq 4n^3\eta^2 L^2 \sum_{r=1}^{n} (D(\boldsymbol{x}^{t-r+1}, \hat{\boldsymbol{x}}^{t-r+1}) + D(\hat{\boldsymbol{x}}^{t-r+1}, \boldsymbol{x}^{t-r})) \tag{A132}$$

$$\leq \frac{1}{2n} \sum_{r=1}^{n} (D(\boldsymbol{x}^{t-r+1}, \hat{\boldsymbol{x}}^{t-r+1}) + D(\hat{\boldsymbol{x}}^{t-r+1}, \boldsymbol{x}^{t-r})). \tag{A133}$$

In the first inequality, we twice used

$$\|\hat{\boldsymbol{x}}^{t+1} - \boldsymbol{x}^t\|_2^2 \leq \langle \nabla h(\hat{\boldsymbol{x}}^{t+1}) - \nabla h(\boldsymbol{x}^t), \hat{\boldsymbol{x}}^{t+1} - \boldsymbol{x}^t \rangle \tag{A134}$$

$$\leq \|\nabla h(\hat{\boldsymbol{x}}^{t+1}) - \nabla h(\boldsymbol{x}^t)\|_2 \|\hat{\boldsymbol{x}}^{t+1} - \boldsymbol{x}^t\|_2. \tag{A135}$$

Here, we used the 1-strong convexity of $h$ and the Cauchy-Schwarz inequality. $\qquad\square$

### E.3 Proof of Lem. A6

*Proof.* For all $\hat{\boldsymbol{x}}, \hat{\boldsymbol{x}}' \in \text{int}(\mathcal{X})$ and all $\boldsymbol{u}, \boldsymbol{u}' \in \mathbb{R}^d$, we define

$$\boldsymbol{x} := \boldsymbol{P}(\hat{\boldsymbol{x}}, \boldsymbol{u}), \quad \boldsymbol{x}' := \boldsymbol{P}(\hat{\boldsymbol{x}}', \boldsymbol{u}'). \tag{A136}$$

We obtain

$$\|\boldsymbol{x} - \boldsymbol{x}'\|_2^2 \overset{\text{a}}{\leq} \langle \nabla h(\boldsymbol{x}) - \nabla h(\boldsymbol{x}'), \boldsymbol{x} - \boldsymbol{x}' \rangle \tag{A137}$$

$$\overset{\text{b}}{=} \langle \nabla h(\hat{\boldsymbol{x}}) - \nabla h(\hat{\boldsymbol{x}}'), \boldsymbol{x} - \boldsymbol{x}' \rangle + \eta \langle \boldsymbol{u} - \boldsymbol{u}', \boldsymbol{x} - \boldsymbol{x}' \rangle \tag{A138}$$

$$\overset{\text{c}}{\leq} (\|\nabla h(\hat{\boldsymbol{x}}) - \nabla h(\hat{\boldsymbol{x}}')\|_2 + \eta \|\boldsymbol{u} - \boldsymbol{u}'\|_2) \|\boldsymbol{x} - \boldsymbol{x}'\|_2. \tag{A139}$$

In (a), we used 1-strong convexity of $h$. In (b), we summed up both the first-order optimality conditions for $\boldsymbol{x}$ and $\boldsymbol{x}'$, i.e.,

$$\langle \eta\boldsymbol{u} - \nabla h(\boldsymbol{x}) + \nabla h(\hat{\boldsymbol{x}}), \boldsymbol{x} - \boldsymbol{x}' \rangle = 0, \quad \langle \eta\boldsymbol{u}' - \nabla h(\boldsymbol{x}') + \nabla h(\hat{\boldsymbol{x}}'), \boldsymbol{x}' - \boldsymbol{x} \rangle = 0. \tag{A140}$$

In (c), we used the Cauchy-Schwarz inequality. In conclusion, we obtain

$$\|\boldsymbol{x} - \boldsymbol{x}'\|_2 \leq \|\nabla h(\hat{\boldsymbol{x}}) - \nabla h(\hat{\boldsymbol{x}}')\|_2 + \eta \|\boldsymbol{u} - \boldsymbol{u}'\|_2. \tag{A141}$$

Here, $\nabla h$ is continuous on $\text{int}(\mathcal{X})$. By the definition of a convex function of Legendre type, $h$ is convex and differentiable. Moreover, $\text{int}(\mathcal{X})$ is an open convex set, and $h(\hat{\boldsymbol{x}})$ is finite in $\hat{\boldsymbol{x}} \in \text{int}(\mathcal{X})$. Thus, Cor. 25.5.1 in the study [44] is applicable, and $h$ is continuously differentiable, meaning that $\nabla h$ is continuous on $\text{int}(\mathcal{X})$. Thus, Eq. (A141) shows that $\boldsymbol{P}$ is continuous on $\text{int}(\mathcal{X}) \times \mathbb{R}^d$. $\qquad\square$

## F    Analysis for Individual Regret

By using Eq. (FO3) for $(\tilde{\mathbf{f}}', \tilde{\mathbf{f}}) = (\tilde{\boldsymbol{x}}_i^t, \tilde{\boldsymbol{x}}_i^{t-1})$, we obtain the "stability" property as

$$\|\boldsymbol{x}_i^t - \boldsymbol{x}_i^{t-1}\|_2 \leq \eta\|\tilde{\boldsymbol{x}}_i^t - \tilde{\boldsymbol{x}}_i^{t-1}\|_2 \tag{A142}$$

$$= \eta\|(m+2)\boldsymbol{u}^{t-m-1} - (m+1)\boldsymbol{u}^{t-m-2}\|_2 \tag{A143}$$

$$\leq \eta\{(m+1)\|\boldsymbol{u}^{t-m-1} - \boldsymbol{u}^{t-m-2}\|_2 + \|\boldsymbol{u}^{t-m-1}\|_2\} \tag{A144}$$

$$\leq (m+2)\eta L. \tag{A145}$$

This leads to the upper bound of individual regret as

$$\mathrm{REG}_i(T) \leq \frac{h_{\max}}{\eta} + \lambda^2\eta\sum_{t=1}^{T}\|\boldsymbol{u}_i^t - \boldsymbol{u}_i^{t-1}\|_2^2 - \frac{1}{4\eta}\sum_{t=1}^{T}\|\boldsymbol{x}_i^t - \boldsymbol{x}_i^{t-1}\|_2^2 \tag{A146}$$

$$\leq \frac{h_{\max}}{\eta} + \lambda^2\eta\sum_{t=1}^{T}\|\boldsymbol{u}^t - \boldsymbol{u}^{t-1}\|_2^2 \tag{A147}$$

$$\leq \frac{h_{\max}}{\eta} + \lambda^2 L^2\eta\sum_{t=1}^{T}\|\boldsymbol{x}^t - \boldsymbol{x}^{t-1}\|_2^2 \tag{A148}$$

$$\leq \frac{h_{\max}}{\eta} + (m+2)^2\lambda^2 N L^4\eta^3 T. \tag{A149}$$

If we set $\eta = 1/(\sqrt{(m+2)\lambda}T^{1/4})$ there, we derive $\mathrm{REG}_i(T) \leq (h_{\max} + NL^4)\sqrt{(m+2)\lambda}T^{1/4} = O(m^{3/2}T^{1/4})$.

## G    Computational Environment

The codes are available at

$$\texttt{https://github.com/CyberAgentAILab/delayed\_learning\_games}$$

The simulations presented in this paper were conducted using the following computational environment.

- Operating System: macOS Monterey (version 12.4)
- Programming Language: Python 3.11.3
- Processor: Apple M1 Pro (10 cores)
- Memory: 32 GB

