# OpenReview forum: "Learning from Delayed Feedback in Games via Extra Prediction"
_NeurIPS.cc/2025/Conference — NeurIPS 2025 poster_

### Official Review · Reviewer_tkVS · 2025-06-12

**Clarity:** 4
**Significance:** 3
**Originality:** 2
**Rating:** 5
**Confidence:** 4

**Summary:**

This paper studies the problem of online learning in games in a deterministic setting. It especially considers a case where the feedback is delayed by a number $m$ of steps at each iteration. In this setting, the constant regret that can be achieved by the Optimistic Follow The Regularized Leader (OFTRL)algorithm is shown theoretically to not be obtainable anymore.

For this reason, the paper proposes the Weighted Optimistic Follow The Regularized Leader (WOFTRL) algorithm that builds on the idea of OFTRL, but with a different weight for the last iterate. The algorithm recovers the constant regret and demonstrates good empirical results.

**Questions:**

What makes obtaining last-iterate convergence hard in this setting? Do you think last-iterate convergence could also be obtained in a stochastic setting (e.g. with each player sampling from their policy instead of the average reward)?

**Ethical Concerns:**

["NO or VERY MINOR ethics concerns only"]

**Final Justification:**

I think the paper is rigorous, the contributions are interesting and the authors mention potential improvement to the results, hence my final grade.

**Limitations:**

Not applicable.

**Paper Formatting Concerns:**

No formatting issues were noticed

**Quality:**

3

**Strengths And Weaknesses:**

Strengths:

- The article is well written and well structured, the theoretical motivations are clear.

- It proposes an (although not extremely novel) algorithm that succeeds where the current literature fails.

- All theoretical findings are proven with some intuitions given in the main paper. The experiments corroborate nicely with the theorem.

Weaknesses:

- The theoretical applications of the result are not obvious, while the practical applications may require a more flexible setting (as especially mentioned in the conclusion).

- A small summary of the contributions could be nice.

---

> ### Author Rebuttal · Authors · 2025-07-30
>
> Thank you for your review and positive evaluation. We are also grateful for your insightful comments, leading our research to the next level. We address your comments one by one in the following.
>
> ---
>
> **Weakness 1)** Theoretical applications:
>
> Although we do not provide a formal proof, our study suggests a potential theoretical application. This study, for the first time, demonstrates the merit of taking the extra ($n$ times) prediction in the aspects of regret and convergence. It provides an idea that increasing the optimistic weight $n$ (with decreasing learning rate $\eta$) may enhance the accuracy of future predictions. This idea could be valuable not only in delayed settings but also in delay-free ones.
>
> ---
>
> **Weakness 2)** Small summary of the contributions:
>
> We will shorten Lines 46-59 as well as possible.
>
> ---
>
> **Question 1)** Extension to last-iterate convergence:
>
> Fortunately, **by further research, we could improve best-iterate convergence to last-iterate convergence**. It was very challenging to find a Lyapunov function. Indeed, the naive idea that the distance from the Nash equilibrium can be a Lyapunov function fails. As shown in Fig.3, due to delayed feedback, the distance from the Nash equilibrium may increase initially. Our manuscript will be updated to last-iterate convergence in the camera-ready phase.
>
> ---
>
> **Question 2)** Extension to stochastic setting:
>
> Several papers have recently been published on such stochastic (e.g., bandit) feedback in learning in games ([a1, a2]), but it seems very difficult to obtain the regret lower bound even if no feedback delay exists. Because bandit feedback requires a different kind of discussion from our full feedback, we leave it as future work.
>
> ---
>
> [a1] Ito, S., Luo, H., Tsuchiya, T., & Wu, Y. (2025). Instance-dependent regret bounds for learning two-player zero-sum games with bandit feedback. arXiv preprint arXiv:2502.17625.
>
> [a2] Fiegel, C., Menard, P., Kozuno, T., Valko, M., & Perchet, V. (2025). The Harder Path: Last Iterate Convergence for Uncoupled Learning in Zero-Sum Games with Bandit Feedback. In ICML.

---

> > ### Comment · Reviewer_tkVS · 2025-08-04
> >
> > Thank you for your response, having a last-iterate convergence result would be a great addition to the paper. I maintain my current score.

---

> > > ### Author Response · Authors · 2025-08-05
> > >
> > > Thank you for your encouragement. We promise to update our result to the last-iterate convergence.

---

### Official Review · Reviewer_ebu8 · 2025-06-27

**Clarity:** 2
**Significance:** 2
**Originality:** 3
**Rating:** 4
**Confidence:** 3

**Summary:**

This paper tackles the challenge of learning in multi‐agent games when there is a delay in observing past rewards. It first shows that even a single‐step feedback delay degrades the performance of the standard Optimistic Follow‐the‐Regularized‐Leader (OFTRL) algorithm, in both regret and convergence. To counteract this, the authors introduce Weighted OFTRL (WOFTRL), which amplifies the optimism in the predicted next‐round reward by a tunable weight. They prove that if this optimistic weight exceeds the length of the delay, WOFTRL restores the desirable properties of OFTRL: it achieves constant (O(1)) regret in general‐sum normal‐form games and ensures best‐iterate convergence to a Nash equilibrium in polymatrix zero‐sum games. Finally, empirical experiments corroborate the theoretical findings, demonstrating that WOFTRL effectively neutralizes the adverse effects of delayed feedback.

**Questions:**

(Q1) Is multi-linear utility common in literature? Can the authors provide examples of pre-existing work that also uses such utility/expected payoff?

(Q2) What if the game has multiple Nash Equilibrium, does the algorithm still holds?

(Q3) For Theorem 6, could there be any other choice of learning rate that improves the regret? How does the regret depend on $\eta$? - I would like to see some discussion about this following Theorem 6.

**Ethical Concerns:**

["NO or VERY MINOR ethics concerns only"]

**Final Justification:**

The authors have addressed my concerns so I raised my score.

**Limitations:**

Yes

**Quality:**

3

**Strengths And Weaknesses:**

Strength:
The paper is in general well written, empirical experiments are provided to demonstrate the effectiveness of the algorithms.

Weaknesses:

(W1) In related literature, the authors mention that "The delays are known to worsen the regret in the full feedback and bandit feedback", but didn't provide detailed discussion about how the regret is worsened. And similarly for the time-varying games literature, how regret and convergence are affected is not sufficiently discussed.

(W2) Some notations are a little confusing, e.g. what's the definition of $h_{\max}$ on line 182?

(W3) The definition 9 introduces RVU property defined from [5], what's the full name of 'RVU', is there any constraint on the $\alpha, \beta, \gamma$ for this definition? --- I'd appreciate it if the authors could include more discussions when introduce this new term.

(W4) Is there any other algorithm in pre-existing work that also shows O(1) regret? If so how does this algorithm compare to them? How do their regret results depend on parameters like $T, m$?

I'm willing to raise my score if my concerns and questions can be addressed.

---

> ### Author Rebuttal · Authors · 2025-07-30
>
> Thank you for carefully reading our paper. Your comments are related to the fundamental (background) knowledge for our contributions. We believe we can address all of them below. In particular, if you recognize the **commonality (Q1), novelty (W4), and strength (Q2, 3) of this work**, we will take you up on your kind offer
>
> - “I'm willing to raise my score if my concerns and questions can be addressed”,
>
> and we hope that you will reassess our paper.
>
> ---
>
> **Weakness 1)** Detailed relation to previous literature:
>
> We just cite Refs. [22-42] to strengthen motivation to consider the problem of delayed feedback in multi-agent learning. In fact, Refs. [22-34] treats **single-agent** learning, which is meaningless to be compared with this study. For example, a delay $m$ is known to worsen regret by $O(\sqrt{mT})$ in Ref. [30], but the baseline regret $O(\sqrt{T})$ is different from ours $O(1)$. The same applies to works on time-varying games—their assumptions differ incomparably from ours.
>
> ---
>
> **Weakness 2)** Definition of $h_{\max}$:
>
> We will note $h_{max} = \sum_{i}\max{x_{i}} h(x_{i})$, thank you.
>
> ---
>
> **Weakness 3)** For the RVU property:
>
> Its full name is the Regret bounded by Variation in Utilities (RVU) property. Its constraint is $\alpha, \beta, \gamma>0$. We will reflect these in the next paper.
>
> ---
>
> **Weakness 4)** For other works showing $O(1)$ regret:
>
> As far as we know, there is **no existing work showing $O(1)$ regret in delayed multi-agent learning**. We believe that in multi-agent learning, delayed full feedback was first proposed by us and solved by weighted prediction. Fortunately, **no other reviewers denied our novelty**.
>
> ---
>
> **Question 1)** For commonality of multi-linear utility function:
>
> The multi-linear utility is **commonly seen in the field of learning in games**. Normal-form games, represented by multi-linear utility functions, are one of the standard research subjects. In fact, many of our references (Refs. [1, 5, 6, 10-15, 35-44]) consider normal-form games and suppose multi-linear utility functions.
>
> ---
>
> **Question 2)** For multiple Nash equilibria:
>
> This study does not assume the uniqueness of the Nash equilibrium, and thus **all the results hold for games with multiple Nash equilibria**.
>
> ---
>
> **Question 3)** For other choices of $\eta$ in Thm. 6:
>
> Sorry for our statement confusing you. **$\eta=O(1/\sqrt{T})$ achieves the smallest regret lower bound $\Omega(\sqrt{T})$ among all possible $\eta$**. Indeed, other choices of $\eta$ enlarges the regret lower bound because ${\rm RegTot}^{2}$ is decomposed into the two terms of $\Omega(1/\eta^2)$ and $\Omega(e^{\eta^2 T}/\eta^{2})$ as shown in the proof.

---

> > ### Author Response · Authors · 2025-08-05
> >
> > We hope that our response has helped you understand. Please do not hesitate to ask any questions.

---

> > > ### Comment · Reviewer_ebu8 · 2025-08-08
> > >
> > > I appreciate your respone and I've already raised my score to 4.

---

### Official Review · Reviewer_E3gP · 2025-06-30

**Clarity:** 4
**Significance:** 3
**Originality:** 3
**Rating:** 4
**Confidence:** 4

**Summary:**

This paper studies the problem of learning in games with time-delayed feedback. In this setting, a learner must choose their next strategy $x_i^{t+1}$ based only on rewards received from rounds $1$ to $t - m$; the rewards from the most recent $m$ rounds are not yet available. The authors focus on variants of Optimistic Follow-The-Regularized-Leader (OFTRL) and analyze both regret and convergence to Nash equilibrium when all players adopt the same learning algorithm.

The paper begins by showing that in matching pennies, if both players use OFTRL and there is a delay of $m \geq 1$, each player suffers a regret of $\Omega(\sqrt{T})$, and their strategies diverge from the equilibrium. To address this, the authors propose a new variant called **Weighted OFTRL (WOFTRL)**. They prove that WOFTRL satisfies the **RVU property** (Regret bounded by Variation in Utilities), which allows them to upper bound total regret by $O(1)$. These theoretical results are supported by experiments on matching pennies and Sato's game.

The paper then extends the analysis to polymatrix games, showing **best-iterate convergence** of WOFTRL to the Nash equilibrium. Additional experiments on weighted Rock-Paper-Scissors demonstrate **last-iterate convergence** when $n > m$, and non-convergence when $n \leq m$.

**Questions:**

1. How robust do you believe the approach of scaling the optimistic term by $m+1$ is? For instance, if the value of $m$ varies from round to round, do you think your method can be generalized to handle such a setting?
2. From a theoretical standpoint, what happens when $n$ is significantly larger than $m+1$? Your experiments show promising results in this regime, but is there any trade-off that may not be immediately apparent?

**Ethical Concerns:**

["NO or VERY MINOR ethics concerns only"]

**Final Justification:**

I went through the discussion between the reviewer wW5Q and the authors and found the response unsatisfactory. I had initially assumed that a social regret of $O(1)$ would naturally imply an individual regret of $O(1)$ as well. However, due to certain nuances in the definition, the $O(1)$ social regret only translates to an individual regret of $O(T^{1/4})$, which now seems less compelling in hindsight.

While I appreciate other aspects of the paper, these concerns have led me to downgrade my score to **"4: Borderline Accept."**

**Limitations:**

No such limitation.

**Quality:**

4

**Strengths And Weaknesses:**

The paper addresses an important time-delayed feedback model and presents a compelling set of results, as outlined in the summary. I particularly appreciate the insightful interpretation of why OFTRL fails to achieve strong guarantees in this setting, and how WOFTRL succeeds. Specifically, the authors frame the behavior of OFTRL and WOFTRL in terms of expansion due to time delay, contraction from optimism, and expansion from accumulated discretization errors. While scaling the optimistic term by $m+1$ may seem like a simple modification, it leads to meaningful new insights into the algorithm’s behavior and is a contribution worthy of recognition.

I did not identify any significant weaknesses. However, there is a small typo on line 69 that should be corrected.

-----

I went through the discussion between the reviewer wW5Q and the authors and found the response unsatisfactory. I had initially assumed that a social regret of $O(1)$ would naturally imply an individual regret of $O(1)$ as well. However, due to certain nuances in the definition, the $O(1)$ social regret only translates to an individual regret of $O(T^{1/4})$, which now seems less compelling in hindsight.

While I appreciate other aspects of the paper, these concerns have led me to downgrade my score to **"4: Borderline Accept."**

---

> ### Author Rebuttal · Authors · 2025-07-30
>
> Thank you for your review and positive evaluation. We further appreciate your insightful questions that advance our research. The following is our answer.
>
> ---
>
> **Question 1)** For stochastic delay $m$:
>
> Thank you for the good point. We have a **simple idea for dealing with stochastic delay**. If we use a larger $n$ than all the possible delays $m$ and refer to the reward $n-1$ steps before (not $m$ steps before), both the $O(1)$-regret and the last-iterate convergence can still be achieved. We will briefly discuss this idea in the Conclusion.
>
> ---
>
> **Question 2)** For trade-off due to significantly larger $n$ than $m+1$:
>
> If you enlarge $n$ while keeping the learning rate $\eta$ fixed, a tradeoff problem arises. Indeed, Eq. (14) shows the pros and cons of taking excessively large $n$. A larger $n$ accelerates convergence via the second term ($O(\eta)$), but also increases prediction error via the third term ($O(\eta^2)$). **If $\eta$ is too large, the prediction error dominates and impedes convergence** (see Lines 217–223). Unfortunately, we have not succeeded in reflecting this tradeoff in regret or convergence rate.

---

> > ### Comment · Reviewer_E3gP · 2025-08-01
> >
> > Thank you for the rebuttal. I am satisfied with your response and will be retaining my current score.

---

> > > ### Author Response · Authors · 2025-08-05
> > >
> > > We are glad you liked our response. Please let me know if you have any further questions.

---

### Official Review · Reviewer_wW5Q · 2025-07-03

**Clarity:** 2
**Significance:** 2
**Originality:** 3
**Rating:** 2
**Confidence:** 4

**Summary:**

This paper analyzes learning dynamics in multiplayer general-sum games where each player only observes a gradient delayed feedback. The authors first show that when using optimistic follow-the-regularized-leader (OFTRL), which is one of the most standard algorithms in the absence of delay in learning in normal-form games, even a small amount of delay prevents the algorithm from achieving fast rates.
In particular, the sum of all players’ regrets (social regret) is lower bounded by $\sqrt{T}$ under a specific setting.
Motivated by this, the authors propose a new algorithm based on a weighted version of OFTRL. This algorithm repeatedly reuses the most recent feedback (more than the constant delay length), and they prove that this technique allows the social regret to be bounded by a constant.
They further support these theoretical findings with numerical experiments and also investigate the case of polymatrix zero-sum games.

**Questions:**

The reviewer expects the authors to respond to the questions in the Strengths and Weaknesses section.

**Ethical Concerns:**

["NO or VERY MINOR ethics concerns only"]

**Final Justification:**

The most concerning aspect of this paper is its dependence on the delay time $m$, which is the most critical parameter in the study.
The upper bound provided is $O(m^2)$, which appears to be far from the optimal dependence.
Due to this weakness, it is difficult to recommend this paper for acceptance for now.

**Limitations:**

The negative social impact does not appear to be relevant to this paper.

**Quality:**

3

**Strengths And Weaknesses:**

The paper is well-written and easy to follow the contents. The writing is generally well-organized, and the problem of learning in games with delayed feedback is an important setting to study. Moreover, the proposed algorithm that achieves the regret upper bound appears sound, and the authors validate its effectiveness through numerical experiments.

That said, I believe there are several points the paper could be improved.

The first weakness of this paper is that the entire analysis focuses solely on the social regret, the sum of the individual regrets of all players. It is unclear why the authors do not consider individual regret bounds for each player.
Related to this, the paper often fails to clearly distinguish whether a statement refers to social regret or individual regret, which leads to confusion. For example:
- In the abstract, it says "regret is constant in general-sum normal-form game", but it is hard to interpret "regret" here as "social regret".
- In Theorem 6, the phrase “they suffer from the regret $\Omega(\sqrt{T})$” gives the misleading impression that each player suffers this regret, whereas the actual lower bound seems to be on the social regret only, as can be seen from the proof.
- For the same reason, the sentence in line 111 is also confusing.
- The definition of the RVU property in Definition 9 is somewhat unnatural. In [5] and many related works, the RVU property is typically defined with respect to each player's regret, and it is not standard to define it directly for social regret.


The second major weakness of this paper is that the the interpretation of the lower bounds in Section 3 is unclear. In Theorem 6, the authors analyze OFTRL under a learning rate of $\eta \simeq 1/\sqrt{T}$, whereas [5] and related works typically use constant learning rates.
Suffering $\sqrt{T}$ regret under this choice of $\eta$ seem natural. Why do authors analyze OFTRL under this particular learning rate?

Another issue with the lower bound is that it assumes an unconstrained setting. In studies like [5], fast convergence results are usually established under bounded feasible sets (such as the probability simplex). However, this paper does not justify why an unconstrained setting is appropriate. Can the authors provide a justification?

The third weakness of this paper is that the regret upper bound derived in this paper does not discuss whether its dependence on the delay parameter $m$ is optimal. Can this dependence be improved? Or can the authors provide a matching lower bound to show the upper bound is tight?

Some of the terminology used in the paper could be more precise. In Algorithm 4, the authors call the algorithm "optimistic FTRL" under the choice $m_i^t = u_i^{t-m}$, but in the literature, "optimistic FTRL" usually refers to the more general case with arbitrary $m_i^t$. Referring to the general case (Definition 2) as "generalized FTRL" is also not appropriate.

Minor comments:
- Line 126: $\mathcal{X}^*$ is undefined.
- Eq. (10): the notation $\tilde{x}_i^t$ is very close in appearance to the notation of actions $x$, which is confusing. The notation used in (A58) also suffers from the same issue.
- The proof has so many typos that needs to be fixed.
- Line 422: "1-convexity" should be corrected to "1-strong convexity."

---

> ### Author Rebuttal · Authors · 2025-07-30
>
> Thank you for your effort and comments. As far as we see, **your comments (W1, 2, 4) mainly concern presentation issues** and do not challenge our core contributions:
> - Identifying that a finite delay $m$ significantly degrades performance in terms of regret and convergence.
> - Proving that an additional prediction restores the good performance.
>
> Your comments are constructive to improve our presentation, and we can **immediately reflect your comments in the next version**. If you have any other concerns, please let us know. The following is our one-by-one responses.
>
> ---
>
> **Weakness 1)** Distinction between individual and social regret:
>
> We appreciate your support. To avoid confusing readers, we will clarify that the regret is social regret. Because prior work in learning in games typically focuses on social regret (e.g., Refs. [5, 6, 35–38]), limiting our **analysis to social regret is not a weakness**.
>
> For the RVU property, we will revise it based on individual regret. This revision does not affect our contribution at all.
>
> ---
>
> **Weakness 2)** For other choices of $\eta$ in Thm. 6
>
> Sorry for our statement confusing you. **$\eta=O(1/\sqrt{T})$ achieves the smallest regret lower bound $\Omega(\sqrt{T})$ among all possible $\eta$**. Indeed, other choices of $\eta$ enlarges the regret lower bound because ${\rm RegTot}^{2}$ is decomposed into the two terms of $\Omega(1/\eta^2)$ and $\Omega(e^{\eta^2 T}/\eta^{2})$ as shown in the proof.
>
> For justification of the unconstrained setting, please reread our remark (Lines 151-156). The unconstrained setting is useful for analyzing the dynamics of the algorithm and for obtaining the regret lower bound because we can avoid considering dynamics on the boundary of the strategy space. Here, the boundary dynamics is no more than an exceptional case for the whole dynamics, and the regret bound is not expected to differ significantly between constrained and unconstrained settings. Indeed, previous works showed that the same phenomenon occurs independently of the constrained or unconstrained setting. For example, convergence in zero-sum games similarly occurs (Refs. [9] (unconstrained) and [10] (constrained)), and convergence in time-varying zero-sum games does (Refs. [40] (unconstrained) and [41] (constrained)). Our simulations also justify this expectation (see Fig. 3). There are **many reasons why the unconstrained setting cannot be a weakness**.
>
> ---
>
> **Weakness 3)** For regret dependence on $m$:
>
> Thank you for the good point. When $n=m+1$, WOFTRL incurs the regret of $O(m^2)$. This $m^2$ arises because we evaluate $m$ true rewards (from time $t-m$ to $t$) using the reward before $m$ steps (see Eq. (A69)-(A73) in the Appendix). Unfortunately, we cannot prove that this dependency is tight.
>
> ---
>
> **Weakness 4)** Terminology of optimistic/generalized FTRL:
>
> For convenience, we name the specific case $m^t = u^{t-m}$ as Optimistic FTRL, following recent literature (e.g., Refs. [12, 40]). To distinguish it from the generalized form of $m^t$, we continue to use "Generalized FTRL" in our paper. However, we will add a footnote noting that this “Generalized” FTRL is often called Optimistic FTRL in several prior works (e.g., Refs. [5, 6]).

---

> > ### Comment · Reviewer_wW5Q · 2025-08-01
> > **Replies to the rebuttal**
> >
> > The reviewer thanks the authors for their rebuttal.
> >
> > > **your comments (W1, 2, 4) mainly concern presentation issues** and do not challenge our core contributions:
> >
> > I would like to respectfully note that both the suggestion to consider individual regret and the discussion regarding the lower bound are not presentation but primarily technical issues. Please find our response to your rebuttal below.
> >
> > > Because prior work in learning in games typically focuses on social regret (e.g., Refs. [5, 6, 35–38]), limiting our **analysis to social regret is not a weakness**.
> >
> > I respectfully disagree with this opinion. In particular, among the references you listed, [5], [6], and [35] appear to also consider individual regret. Could you please clarify your position on this point?
> > In general, upper bounding the social regret is considered to be relatively easier than upper bounding the individual regret.
> >
> > > **Weakness 2)** For other choices of  in Thm. 6 … For justification of the unconstrained setting, please reread our remark (Lines 151-156).
> > … and the regret bound is not expected to differ significantly between constrained and unconstrained settings…..
> >
> > The explanation given in the remark on Lines 151–156 appears quite insufficient to justify the reason for considering the unconstrained setting.
> >
> > For the comments in the rebuttal, the authors mention that the regret does not differ significantly between the constrained and unconstrained settings. However, is there any justification for this claim? The fact that similar regret upper bounds (or convergence rate) are obtained in, for example, references [9, 10] does not necessarily imply that they have the same property in nature, and does not imply that the same would hold in the presence of delayed feedback. Could the authors provide more concrete justification for why a similar result should be expected in the delayed feedback setting? Is it difficult to show an $\Omega(\sqrt{T})$ bound in an unconstrained setting?
> >
> > Additionally, the choice of step size $\eta = 1/\sqrt{T}$ seems to differ from the one that is used to establish your regret upper bound. Can the authors clarify this point?

---

> ### Author Response · Authors · 2025-08-05
> **Thank you for your reply**
>
> **Further Q1)** Individual regret
>
> **[Apology for your confusion]** First, we are really sorry for confusing you. We referred to the papers because we wanted to show that social regret alone is a sufficient result (indeed, some papers analyze social regret alone) and that social regret is prioritized over individual one. Anyway, our response, “prior work in learning in games typically focuses on social regret,” sounds like an overclaim. We are sorry again.
>
> **[Analysis of individual regret]** Although we still do not understand why the reviewer is committed to individual regret, we sincerely respect your comments and promise to add the results for individual regret with time delays in the camera-ready phase. We will prove that even under delayed feedback, WOFTRL achieves $O(T^{1/4})$ individual regret, following the methodology of Theorem 11 and Corollary 12 in Ref. [5]. The proof will be very short, and only minor revisions will be required to the manuscript. This is because we have already obtained the RVU property as the main result of this study (but with different $\alpha, \beta, \gamma$), and a similar result to Theorem 11 will be immediately obtained. In fact, Ref. [5] devotes very few sentences to proving Theorem 11 and Corollary 12. Keeping in mind that our paper will include the results for individual regret in the next version, could you reevaluate this paper, please?
>
> ***
>
> **Further Q2)** Step size of $\eta$
>
> **[Difference choice of $\eta$]** The different choice of $\eta$ between Thm. 6 and Cor. 11 is no problem. In Thm. 6, we set $\eta=O(1/\sqrt{T})$ in OFTRL, which suffers $\Omega(\sqrt{T})$ regret as the lower bound. In Cor. 11, we set $\eta=O(1)$ in WOFTRL, which achieves $O(1)$ regret as the upper bound. Because Thm. 6 and Cor. 11 have different algorithms and purposes, they do not have to be discussed with the same $\eta$.
>
> **[$\eta=O(1)$ in Thm. 6]** We can consider the step size of $\eta=O(1)$ in Thm. 6 according to Cor. 11. Then, social regret becomes ${\rm RegTot}=\Omega(\exp(\frac{1}{2}\alpha T))$, unfortunately exceeding $\Omega(\sqrt{T})$ which is obtained by $\eta=O(1/\sqrt{T})$. Thus, the lower bound $\Omega(\sqrt{T})$ by $\eta=O(1/\sqrt{T})$ should be exhibited in our Theorem 6.
>
> ***
>
> **Further Q3)** For constrained setting
>
> **[Difficulty in analyzing constrained setting]** In the proof, we have shown that the learning dynamics cycle around the Nash equilibrium with the radius expanding exponentially (Lemma A1). Such divergent dynamics are also observed in FTRL, leading to $O(\sqrt{T})$ regret. In the constrained setting, the divergent dynamics reach the boundary of the strategy space and continue to cycle on the boundary. It is difficult to analyze the regret lower bound in the specific dynamics on such boundaries.
>
> **[Justification for similarity between constraint and unconstraint]** As explained in the above, the divergent dynamics (seen in FTRL) contribute to $\Omega(\sqrt{T})$-regret. The timing when the dynamics reach the boundary in the constrained setting is after they have deviated sufficiently away from the Nash equilibrium. Thus, the divergent dynamics themselves are similar between the constrained and unconstrained settings. Hence, we conclude that the $\Omega(\sqrt{T})$ is expected to be obtained also in the constrained setting.
>
> ***
>
> **Conclusion)**
>
> Again, we really appreciate your valuable time and reply. At least, the unconstrained setting is only related to our negative results (Thms. 6 and 7) and irrelevant to our positive contributions (Thm. 10 and Cors. 11 and 13). In addition, analyzing individual regret often provides only a small step forward from social regret (indeed, we could easily add the results for individual regret). Instead of individual regret, we continue to believe that we should highlight the convergence to the equilibrium, which usually requires a different kind of discussion. We would like you to reconsider how serious the technical flaws you raised are in this work.

---

> ### Comment · Reviewer_wW5Q · 2025-08-07
>
> Thank you to the authors for their response. I understood the clarification regarding Theorem 6.
>
> > WOFTRL achieves  $O(T^{1/4})$ individual regret, following the methodology of Theorem 11 and Corollary 12 in Ref. [5].
>
> I feel that an individual regret bound of $O(T^{1/4})$ is quite far from optimal.
> For example, Farina et al. (2022) show that in the non-delayed setting, the individual external regret can be bounded by $O(\mathrm{poly} \log T)$.
> It seems like a significant compromise to allow the individual regret to be as large as $O(T^{1/4})$ just because the setting involves delays.
>
> [Farina et al. 2022] Kernelized Multiplicative Weights for 0/1-Polyhedral Games: Bridging the Gap
> Between Learning in Extensive-Form and Normal-Form Games, ICML 2022
>
> Furthermore, regarding the dependency on the delay time $m$, which is the most important parameter in this paper, I believe that the $O(m^2)$ dependency is far from optimal, as mentioned in the review. It is quite undesirable that the main results of the paper, specifically Theorem 10 and Corollary 11, exhibit such a suboptimal dependency. If the dependence on such a crucial parameter appears to be highly sub-optimal, then in order to meet the NeurIPS bar, it may be desirable that the paper either establish a matching lower bound for that dependency, or present other compelling extensions such as a generalization to the variable delay setting.
>
> As a minor point, since the delay time $m$ is the most important parameter in this work, the bound in Corollary 11 should be written as $O(m^2)$ rather than simply $O(1)$.
>
> I have no further questions, but I will update my score after discussing with the other reviewers if needed.

---

> ### Author Response · Authors · 2025-08-08
>
> **We are grateful that your questions were resolved through our discussion.** We kindly hope you will reflect this in your next assessment.
>
> ***
>
> This study has yielded several surprising results, independent of the quantitative analysis related to the tight bound. Indeed, we found, for the first time, that OFTRL does not perform well under delayed full feedback. Since OFTRL is one of the most well-known and widely applied algorithms, it is surprising that even a minimal delay of $m = 1$ qualitatively degrades its performance---bringing it down to a level comparable to FTRL. Furthermore, our proposed algorithm, WOFTRL, successfully restores the strong performance in terms of regret and convergence that OFTRL originally possessed. **We believe that this qualitatively novel problem and solution meet the NeurIPS standards.**
>
> ***
>
> We understand that you believe our regret bounds are far from optimal when compared to previous studies conducted under the **non-delayed setting**. First, we emphasize that the $O(m^{2})$ social regret is not excessively loose because the accumulated error due to delayed rewards is on the order of $m \times m$, as mentioned earlier. Second, note that results from the non-delayed setting do not necessarily carry over to the delayed feedback setting, as you also acknowledged in your previous reply. **A fact is that for delayed feedback, $O(m^{2})$ social regret and $O(T^{1/4})$ individual regret by WOFTRL are the best in the current world.**
>
> ***
>
> Anyway, we greatly appreciate your comments and the fruitful discussion. Thanks to your feedback, we could extend our results to include individual regret, which we feel is meaningful to some extent.

---

### Author Response · Authors · 2025-08-09
**Summary of discussion phase**

We appreciate all the reviewers’ efforts and their thoughtful comments. Thanks to the valuable discussions with them, we believe that, in addition to defending our main contributions, we were able to further improve our work.

***

**Reviewer wW5Q** raised questions regarding the result of Theorem 6, which we were able to successfully address during the discussion. Following the reviewer’s suggestion, we additionally derived the individual regret bound. Although the reviewer may still have concerns that our regret bound may not be tight, we believe that our contributions are satisfactory because **we proposed and qualitatively resolved the problem of delayed full feedback for the first time**. The quantitative refinement of the regret bound—while important—can be considered future work.

***

**Reviewer ebu8** raised questions relevant to the basis of our study. We shared the commonality, novelty, and strength of this work with him, which resulted in their positive assessment.

***

**Reviewers E3gP** and **tkVS** highly evaluate our paper. They well understand the content and value of this work and give us insightful suggestions to improve our contributions. We did our best to reflect their suggestions in the manuscript. In particular, we should remark that stochastic delay (by Reviewer E3gP) can be solved by setting a sufficiently large optimistic weight $n$ for delay $n-1$. Also, best-iterate convergence in the present manuscript will be improved to last-iterate convergence (by Reviewer tkVS), which had already been attempted after submission.

***

We thank all the reviewers once again. **The required revisions are not heavy, and we are already ready to submit the manuscript for camera-ready.**

---

### Decision · Program_Chairs · 2025-09-17

**Decision:**

Accept (poster)

**Comment:**

This paper discusses “the problem of time-delayed feedback in learning in games.”  For the most part, the reviewers found the paper to address an important time-delayed feedback model that is rigorous, well-written, and easy to follow.  The results are compelling.  However, there were some concerns, in particular with the paper’s “dependence on the delay time $m$” (Reviewer wW5Q).

In reviewing the discussion about this concern, reviewer E3gP being dissatisfied with the authors’ response (so they dropped their score from accept to borderline accept — see their final justification).  On the other hand, reviewer ebu8 was satisfied with the authors’ response, so they raised their score to 4.  Overall, the reviewers were happy with the paper.

I encourage the authors to thoroughly consider the the concern expressed by reviewer wW5Q in their revision of the paper.